# CHARMM-GUI Multicomponent Assembler for modeling and simulation of complex multicomponent systems

Nathan R. Kern [1], Jumin Lee [2], Yeol Kyo Choi [2] & Wonpil Im [1,2,3] ✉

Atomic-scale molecular modeling and simulation are powerful tools for computational biology. However, constructing models with large, densely packed molecules, non-water solvents, or with combinations of multiple biomembranes, polymers, and nanomaterials remains challenging and requires significant time and expertise. Furthermore, existing tools do not support such assemblies under the periodic boundary conditions (PBC) necessary for molecular simulation. Here, we describe Multicomponent Assembler in CHARMM-GUI that automates complex molecular assembly and simulation input preparation under the PBC. In this work, we demonstrate its versatility by preparing 6 challenging systems with varying density of large components: (1) solvated proteins, (2) solvated proteins with a pre-equilibrated membrane, (3) solvated proteins with a sheet-like nanomaterial, (4) solvated proteins with a sheet-like polymer, (5) a mixed membrane-nanomaterial system, and (6) a sheet-like polymer with gaseous solvent. Multicomponent Assembler is expected to be a unique cyberinfrastructure to study complex interactions between small molecules, biomacromolecules, polymers, and nanomaterials.

Molecular dynamics (MD) simulation becomes essential to study diverse molecular phenomena at atomic resolutions. Advances in computational power and algorithms have enabled simulation models with atom counts in the range of 100 million to several billion, including studies of metal nucleation and sliding[1–4], bacterial cytoplasm[5–7], eukaryotic gene loci[8], synaptic bouton[9], and viral capsids[10,11]. Preparing MD simulation systems typically requires determining the model size and composition, solving a challenging packing problem, and having topological information of each molecule and associated force field (FF) together with simulation input parameters. However, each of these steps presents a challenging barrier for researchers attempting to enter the field, including many experimental researchers who seek to supplement their studies with molecular modeling and simulation.

Many software applications have been developed to facilitate various steps of atomistic model preparation, including FFParam[12], FFTK[13], SwissParam[14], Antechamber[15], CGenFF[16–18], MATCH[19], OpenFF[20], and CHARMM-GUI Ligand Reader & Modeler[21] for FF preparation; PACKMOL[22–24], cellPACK[10,11], TS2CG[25], LipidWrapper[26], and Soup[27] for molecular packing; polyply[28], pysimm[29], and Polymer Builder[30] for building and assembling long polymer chains; and FF-Converter[31,32] and ParmEd[33] for preparing and converting inputs for several simulation programs. Although all software packages require familiarity with the fundamentals of molecular modeling, there are still opportunities to lower the entry barriers for modeling systems with multiple proteins and/or metabolites of interest at the all-atom resolution. Certain common molecular configurations are already handled by simulation and analysis programs directly, such as AmberTools[15], GROMACS[34,35], NAMD/VMD[36,37], and OpenMM[38], which can add a box of water and ions to a single molecule of interest or to a pre-arranged molecular complex or mixture. Similarly, several tools exist to facilitate building combined protein-membrane models by using the template copying method for water placement and by using either a membrane template scheme for positioning lipids or solving the packing problem for the membrane-

[1]Department of Computer Science & Engineering, Lehigh University, Bethlehem, PA, USA. [2]Department of Biological Sciences, Lehigh University, Bethlehem, PA, USA. [3]Department of Bioengineering, Lehigh University, Bethlehem, PA, USA. ✉e-mail: wonpil@lehigh.edu

protein only[25,39–44]. In each of these cases, only a single protein or pre-oriented protein complex can be handled per molecular model. Furthermore, while PACKMOL, cellPACK, and LipidWrapper can pack a wide variety of molecules into nearly any defined shape, neither program currently handles the periodic boundary conditions (PBC) necessary for MD simulations. In addition, significant post-processing is typically required to use the output of these tools to prepare inputs for different simulation programs. CHARMM-GUI[45] is a cyberinfrastructure that guides researchers through building simulation-ready atomistic and coarse-grained models containing a single component or complex of interest in solution or membrane environments. Recent developments have also extended its functionality to cover various nanomaterials and polymer models[30,46].

In this work, we present CHARMM-GUI Multicomponent Assembler (MCA) that solves a complex packing problem for many components of interest under the PBC, enables using pre-equilibrated membrane or membrane-like (nanomaterials and polymer) materials, and generalizes the template-based solvent building approach to cover non-water and mixed solvents. To illustrate MCA's versatility for heterogeneous system building, we prepare and simulate 6 challenging systems at various densities, for a total of 20 systems, using only components generated by other CHARMM-GUI modules: (1) solvated proteins, (2) solvated proteins with a pre-equilibrated membrane, (3) solvated proteins with a sheet-like nanomaterial, (4) solvated proteins with a sheet-like polymer, (5) a mixed membrane-nanomaterial system, and (6) a sheet-like polymer with gaseous solvent.

## Results

### Workflow of Multicomponent Assembler

MCA handles molecular components by grouping them into 5 categories (Table 1) that differ by their general positioning requirements and assumptions. MCA combines components into 6 overall molecular configurations (Fig. 1) using the workflow described below and in Fig. 2. Note that MCA currently uses the CHARMM36(m) FF[47] for protein and lipid components, CGenFF[16–18,48,49] for polymers, and INTER-FACE FF[50–55] for nanomaterials.

**STEP 1 – Read structures and FFs.** Because CHARMM-GUI uses CHARMM for model building and manipulation, it is required to provide molecular structures in both CHARMM protein structure file (PSF) and coordinate (CRD) formats. Additionally, for periodic (nanomaterial) structures with image bonds (between the primary system and the PBC systems), listing the bonds in a CHARMM-formatted image PSF file is necessary. Molecules containing residues already present in the CHARMM and INTERFACE FFs are recognized automatically, but any additional residue topologies and FFs must be provided in CHARMM's residue topology file (RTF) and parameter (PRM) formats with unique identifiers that do not conflict with the existing FFs. Note that these files can be obtained from other CHARMM-GUI modules such as PDB Reader & Modeler[56], Glycan Reader & Modeler[57], Ligand Reader & Modeler[21], Nanomaterial Modeler[46], and Polymer Builder[30].

For each uploaded component, MCA uses CHARMM internal functions to determine the molecular dimensions, volume, solvent-accessible volume, mass, charge, number of residues, and radius of gyration. To facilitate membrane building, the uploaded coordinates are used to determine the volume and center of mass of the molecular regions that would be located within, above, and below a membrane centered at $Z = 0$ with a hydrophobic thickness of 24 Å. The component's dimensions are calculated as its bounding box when the component's primary, secondary, and tertiary axes are aligned with the $X$, $Y$, and $Z$ axes, respectively. The component's length is determined as the maximum distance between any two atoms within the structure. This value is needed to estimate the minimum box size that prevents any component from interacting with its own PBC images. The molecular volume is calculated by polling coordinates using a grid spacing of 0.5 Å within any atom's van der Waals radius, with a maximum probe range of 6 Å, and counting the empty holes within a molecule toward its volume. For solvent-accessible volume, the procedure is repeated with atomic radii increased by 1.4 Å, corresponding to the water radius.

To ensure that segment identifiers are both unique and meaningful, all segments from each input structure are written to separate PDB files to determine the segment type (protein, DNA, RNA, heterogen, carbohydrate, or water). Each input segment is then re-written to a PSF such that the first letter corresponds to the segment type (protein: P, DNA: D, RNA: R, heterogen: H, carbohydrate: C, water: W), and the next two letters designate a unique ID. Thus, the first input protein segment is renamed to PAA, the 27th to PBA, etc. A human-readable map of input-to-output segment IDs is written to a file (rename_map.txt) for the user's convenience.

**STEP 2 – Determine system size and positional constraints, and pack solutes.** To properly specify the components in a given system, users must identify the type of each component as listed in Table 1. If there are no membrane-embedded components, users must specify whether a new membrane should be generated.

In the simplest case, the user already knows the exact number of copies of each component and system dimensions. However, commonly only the relative ratio of components with respect to each other is known. Similarly, instead of knowing the exact system dimensions, the user may know only the fraction of available volume that should be occupied by uploaded components (not including solvent and ions that are handled later). To guide in determining an appropriate system size and number of components, one can specify the relative component ratios, and specify either exact system dimensions or volume fraction and an approximation of the other quantity, as shown in Fig. 3. MCA recommends the closest match to the approximated quantity with the following constraints: (1) no dimension is small enough that a component can interact with its own PBC images, and (2) the volume fraction is low enough that packing can plausibly succeed. To quickly estimate an upper bound on the maximum packing density, we rely on a heuristic that empty space should be no less than the total solvent accessible volume of all components. However, in our observation, achieving volume fractions higher than 30% usually requires

## Table 1 | Supported component types

| Type | Definition | Examples |
|------|-----------|----------|
| Solvated | Large components surrounded by water or vacuum | Protein, protein complex, nanoparticle |
| Solvent | Small, uncharged molecule type in the solvent region | Ethanol, toluene, carbon dioxide, custom water-model |
| Ion | Small, charged molecules within the solvent region | Phenoxide-charged amino acid |
| Membrane-Embedded | Membrane-embedded molecule or complex; must be pre-aligned to a membrane parallel to XY plane whose center is at $Z = 0$ | Membrane protein/complex, nano-tube pore |
| XY Periodic | Any component whose area pre-defines the $X$ and $Y$ system dimensions; if the system contains multiple XY periodic components, they must all have the same $XY$ dimensions | Pre-equilibrated membrane, nanosheet |

Each uploaded component must have a component type based on its positioning and packing requirements.

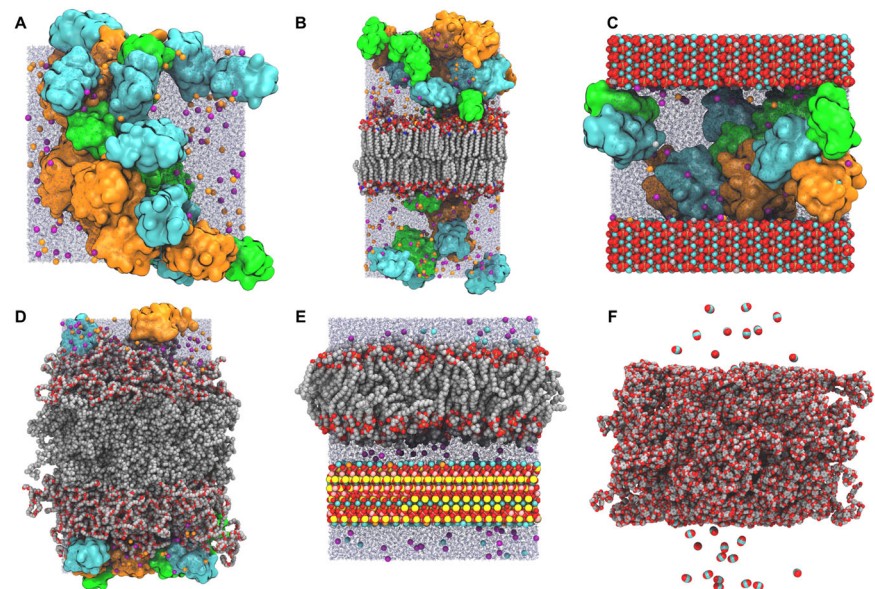

**Fig. 1 | Example system configurations generated by Multicomponent Assembler.** In all figures containing protein, ubiquitin (PDB: 1UBQ) is orange, villin (PDB: 1VII) is green, and protein G (PDB: 3GB1) is cyan. $K^+$ and $Cl^-$ ions are shown as orange and purple spheres, respectively, and water molecules as blue translucent lines. When included, KCl concentration is 0.15 M. **A** Proteins with water and ions only. **B** Pre-equilibrated axolemma membrane combined with proteins. **C** Proteins with a hydroxyapatite slab centered on the unit cell's Z boundary. **D** Artificial membrane of polyethylene oxide-poly(ethylethylene) ($EO_{40}EE_{37}$) with proteins. **E** Supported lipid bilayer consisting of 1-palmitoyl-2-oleoyl-phosphatidylcholine (POPC) membrane and mica separated by a 20 Å water layer. **F** $CO_2$ adsorption on a polyethylene terephthalate (PET) membrane.

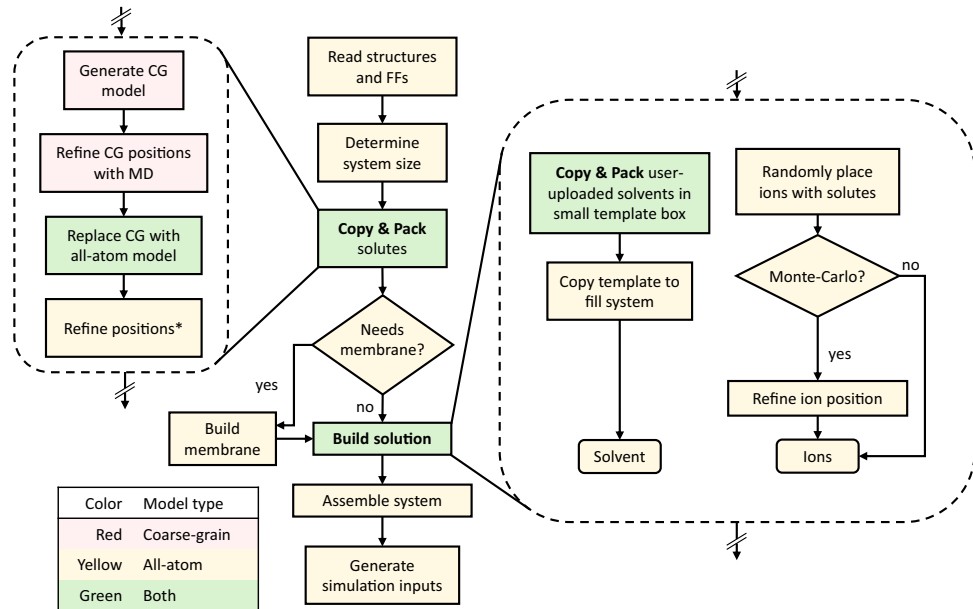

**Fig. 2 | Multicomponent Assembler workflow.** Step colors indicate which model types they use. The Copy & Pack procedure is used to place solutes and to create a small solvent template. CG: coarse-grained. MD: molecular dynamics. *Positions are refined using the algorithm in Supplementary Algorithm 1.

significant trial and error. As shown in our benchmark study (see Supplementary Fig. 1 and Comparison with other programs), the difficulty of finding a collision-free packing depends on the shape of the components and the desired system density. In particular, for high volume fractions (e.g., > 30%), as shown in our benchmark testing, a trial-and-error process is unavoidable in the current approach.

Systems without a membrane or periodic component are modeled with a cubic crystal lattice (X = Y = Z, 90° angles); those with a membrane are tetragonal (X = Y ≠ Z, 90° angles) and require the system width (X or Y) and height (Z) to be determined separately (see Supplementary Fig. 2a); those with a periodic component are orthorhombic (X ≠ Y ≠ Z, 90° angles), with system X and Y dimensions defined by the periodic component's dimensions. For the purpose of estimating the available space for packing, the approximate membrane or periodic component's thickness along Z must be provided, if applicable (see Supplementary Fig. 2b). Furthermore, the provided membrane or periodic component's *XYZ* dimensions are reserved for that component. Note that their volumes are subtracted from the available

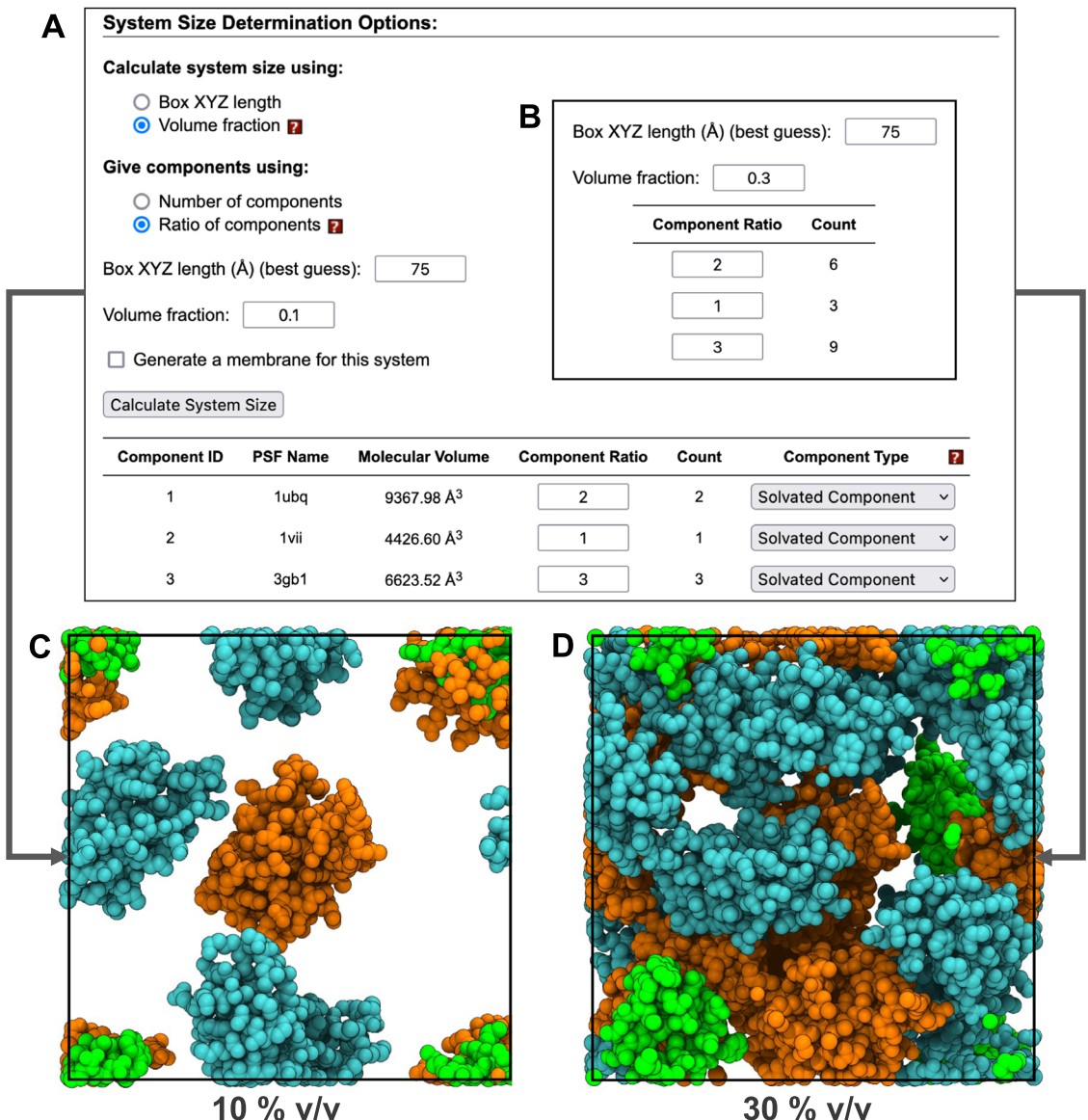

**Fig. 3 | User interface and corresponding packing results with varied volume fraction and component ratios.** 10 % v/v is used in (**A**) and 30 % v/v in (**B**). Both parameters result in a cube with a side length of 74.19 Å. The resulting component count is updated after clicking "Calculate System Size". Packing results of (**A**) and (**B**) are shown in (**C**) and (**D**), respectively. Molecules may cross the system boundaries only if collisions with periodic image atoms are avoided. In (**C**) and (**D**), all atoms are wrapped to the primary cell to illustrate the use of available space. Ubiquitin (PDB: 1UBQ), villin (PDB: 1VII), and protein G (PDB: 3GB1) are colored orange, green, and cyan, respectively.

space for system density calculations and solvent/solute components are prevented from being initialized within or entering the reserved regions.

After the system dimensions and component counts are determined, the user can specify positional constraints (Fig. 4) whose options differ slightly between component types. For solvated components, the options are: (1) "none", which accepts any collision-free position and orientation, (2) "planar (Z) restraint", which fixes the center of mass (COM) of the component to a given Z position, but allows translation along X and Y, and allows any rotation about the component's COM, and (3) "fixed XYZ", in which the component's coordinates are translated so that its COM lies at a given coordinate, and no further movement is allowed. For membrane-embedded components, the options are (1) "none", in which the component's Z positions remain unchanged from their uploaded values, but the component may translate in the X and Y directions and rotate about the Z axis, (2) "planar (Z) restraint", which is like "none", but the Z position may be changed a bit from its uploaded value, and (3) "fixed XY", in which the Z position remains at its initial value, but the user can specify a fixed X, and Y coordinate for the component's COM. Finally, for XY periodic components, the user can specify a COM-Z position since the X and Y lengths of periodic components already define the system's length in X and Y. If no options are selected, the largest component is fixed at the system center. In all cases, the default constraint type is "none".

The non-solvent components are first packed using an uncharged coarse-grained (CG) sphere model that uses 1 to 3 large atoms per copy of each component (Fig. 5). Positions are initialized according to the aforementioned positioning constraints chosen by the user. The CG model is then simulated in 5 iterations with Langevin dynamics for 100 ps using a 2 fs timestep at 500 K with a switching function applied to nonbonded forces at 10 Å plus the largest sphere's radius of gyration. Each iteration of dynamics uses a system size decreasing from 150% of the user's target to 100%. Solvated components are initially

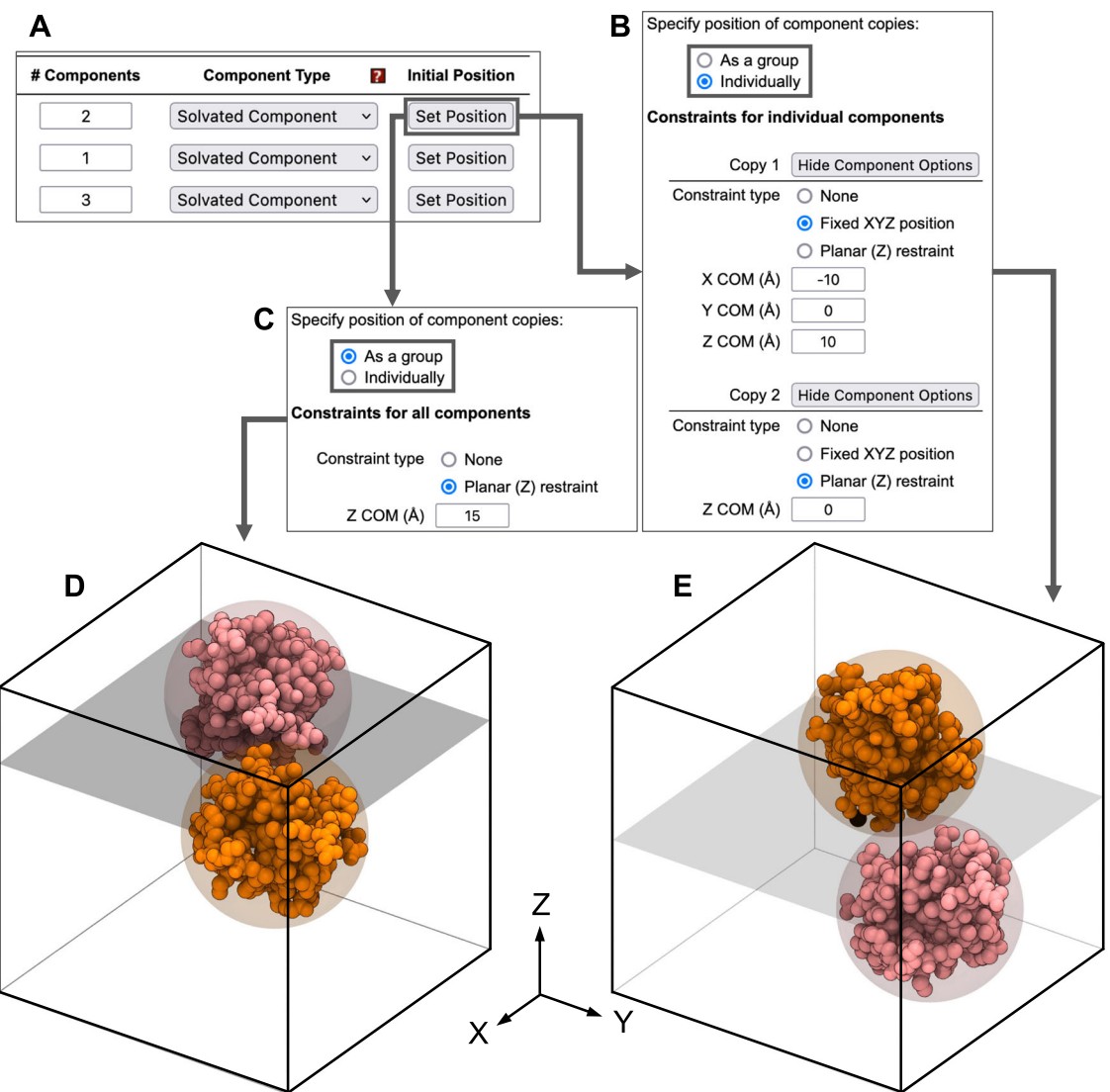

**Fig. 4 | User interface of positioning options for solvated components.** Any component can have its positioning constraints modified by clicking "Set Position", as shown in (**A**). Exact positions can be specified for (**B**) individual components' center of mass (COM), or (**C**) identical planar restraints for all copies. Results of applying the settings in (**C**) and (**B**) are shown in (**D**) and (**E**), respectively. A translucent plane shows the location of each planar restraint, which is applied to both component copies in (**D**), and only the pink copy in (**E**). Corresponding coarse-grained (CG) particles are shown as translucent spheres.

positioned by applying a random rotation and aligning the component's COM with the center of its corresponding CG particle. For membrane components, a root-mean-squared best-fit alignment is performed between the resulting CG coordinates and the reference coordinates calculated in STEP 1; the $Z$ axis rotation and X/Y translation results of this best fit are applied to the all-atom model. Then, collisions are minimized with up to 7 iterations of the greedy conformation search (see Supplementary Algorithm 1).

If packing fails to find a collision-free configuration, MCA stops and emits an error. Otherwise, segments are renamed using the map from STEP 1. Individual copies of a component are distinguished by appending a copy ID to the segment ID, so that the first protein segment's copy becomes PAA1, etc.

**STEP 3 – Build membrane (membrane systems only).** If the system contains membrane-embedded components or the user chooses to generate a membrane, the membrane lipid composition is determined using Membrane Builder[40,41,58] with the system dimensions determined in STEP 2. The lipid packing and replacement procedures are the same as in Membrane Builder, except that the system dimensions during lipid

packing are decreased in 4 iterations from 150% to 120% of target membrane width, and with 21 iterations from 120% to 100%. All membrane lipids built by Membrane Builder are given the segment ID MEMB.

**STEP 4 – Build solution.** The water and ion building procedures in this study follow the same protocols as in Solution Builder[31,32,45], with the additional ability to utilize user-uploaded ions. Users can build any combination of ions supported by the CHARMM36 FF, containing up to 7 atoms, in addition to uploading custom ions. Solvent options allow users to specify either the concentration of any solvent, including water, or a solvent density and ratio of solvent volumes. The solvent options page estimates the number of each generated solvent and ion with the user's settings, although the exact number may differ as described below in STEP 5.

To maintain the approximate solvent ratios, user-uploaded solvent molecules are initially packed into a small box whose size is chosen based on heuristics. The packing procedure follows the same protocols described in STEP 2. After packing, the box is replicated to reach the target box size, and any molecules outside the system boundaries are deleted.

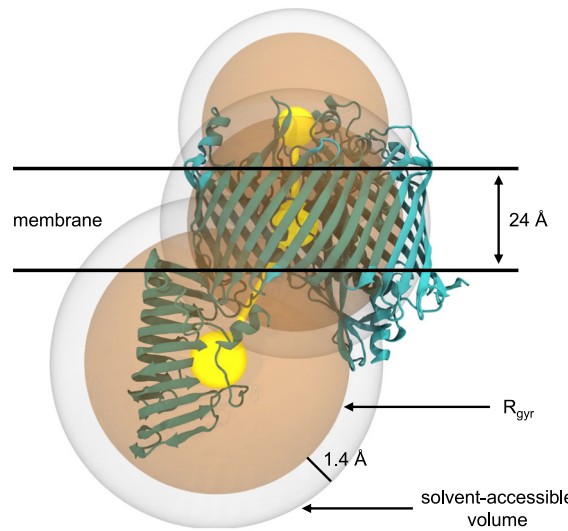

membrane

24 Å

$R_{gyr}$

1.4 Å

solvent-accessible volume

LptD/E: LPS transporter complex (partial)

**Fig. 5 | Coarse-grained representations of molecular components.** For solvated components, a single particle represents the whole component; for membrane-embedded components, 1–3 particles are used, corresponding to the above-membrane, intra-membrane, and below-membrane regions of the component in a 24 Å thick implicit membrane region centered at $Z = 0$. The van der Waals radius of each particle is set to the radius of gyration ($R_g$) of the atoms contained in the corresponding above/intra/below membrane region or omitted if there are no atoms in that region. The solvent accessible volume around a particle corresponds to its molecular volume plus a 1.4 Å thick shell around that particle's volume. LPS: lipopolysaccharide. LptD/E: Lipopolysaccharide transporter protein subunits D and E.

**STEP 5 – Assemble system.** The solvent and non-solvent components are combined by superimposing the solvents with the non-solvents and deleting any solvent molecules that collide with non-solvent molecules within 2.8 Å. If the system contains a membrane or XY periodic component, solvents located in the component's interior (as defined by the user in STEP 2) are also deleted. If a solvent is present in a higher concentration than the user's target, extras are deleted randomly as needed. However, extra solvent molecules are not created to fill space when system assembly results in fewer than the target number of molecules. Finally, any uploaded components with the segment ID MEMB and membrane generated by Membrane Builder (if any) are joined into a single segment, so that appropriate restraints can be generated in STEP 6.

**STEP 6 – Generate simulation inputs.** MCA uses FF-Converter to generate simulation inputs for various MD engines[31,32]. For systems containing periodic nanomaterials or polymers, the generated NVT (constant number of particles, volume, and temperature) equilibration inputs use a multi-step scheme where protein restraints are progressively released. Inputs generated for NPT (constant number of particles, pressure, and temperature) production use anisotropic pressure coupling, and NPAT (constant number of particles, pressure, membrane area, and temperature) production inputs use semi-isotropic pressure coupling with pressure applied only along the $Z$ dimension.

## Protein-protein and protein-membrane contacts

The effects of protein crowding near surfaces are important in biology and industry. The prevalence of membrane surfaces in cells suggests that even cytosolic proteins must interact with membranes, especially in organelles with high surface area such as the endoplasmic reticulum and mitochondria. In laboratory settings, solvated proteins can bind to their container surfaces and leave residues that require strong acids to remove. Nawrocki et al.[59] recently reported the contacts between

crowded proteins and a membrane of cholesterol (CHL1), 1-palmitoyl-2-oleoyl-phosphocholine (POPC), and palmitoyl-sphingomyelin (PSM) via atomistic MD simulation. Protein crowding is difficult to model in atomic details due to the high possibilities for collisions between atoms that cannot be resolved by energy minimization. To study how different surfaces affect protein contact behaviors in crowded environments and to demonstrate the versatility of MCA's modeling capabilities, we reproduced each of the models in Nawrocki et al. and modeled 3 additional membrane types with varying protein concentrations (Fig. 1a–d).

For the models containing proteins and a membrane-like component (see "Methods"), we measured the contact probabilities ($p$) between all pairs of components and aggregated them by component type (Fig. 6). Our results show that protein contact preferences depend greatly on the membrane environment and protein volume fraction. At 5% v/v, proteins are very likely ($p > 0.5$) to contact hydroxyapatite (HAP) and polyethylene oxide-poly(ethylethylene) ($EO_{40}EE_{37}$) but unlikely ($p < 0.1$) to contact CHL1/POPC/PSM membranes. Only ubiquitin and villin show high contact probabilities near the axolemma membrane at 5% v/v. For all membrane types, increasing protein v/v has the effect of increasing the relative proportion of contacts between proteins and other proteins, though contacts between protein and membrane remain frequent ($p > 0.14$) for all protein types and membranes except CHL1/POPC/PSM. Indeed, all protein-CHL1/POPC/PSM contacts surprisingly decrease between 5–10% v/v and increase between 10–30% v/v, suggesting a strong preference for protein-protein contacts that is only overcome by increased protein crowding. The higher protein-membrane contacts at 5% v/v can be explained by the lack of opportunities for protein cluster formation[59]. It should be noted that this trend has a small magnitude: even the highest protein-CHL1/POPC/PSM contact fraction (7.6% for ubiquitin-membrane at 30% v/v) is lower than the lowest protein-membrane contact fraction with other membranes (14.5% for protein G-axolemma at 5% v/v). We believe more simulation replicas are required to establish the statistical significance of this trend reversal. The overall low protein-CHL1/POPC/PSM contacts and high contacts with the other membranes we simulated are consistent with Nawrocki et al.'s[59] finding that proteins are thermodynamically excluded from membranes with no charged lipids.

Proteins uniformly prefer contacts with the same type of protein than with other protein types, with two notable exceptions. First, at 5% protein v/v, villin prefers contacts with axolemma membrane over contacts with other copies of villin. Second, no obvious contact patterns between proteins are observed near HAP for protein v/v < 30% because contacts with HAP are strongly preferred.

Visualization of the HAP and $EO_{40}EE_{37}$ simulation trajectories suggests different reasons for their frequent protein contacts. The duration of contact events between proteins and HAP is higher than that in other membranes, with several contact events lasting longer than 100 ns (Supplemental Fig. 3). In contrast, the hydrophilic EO domain of $EO_{40}EE_{37}$ dissolved so much into water throughout simulation that there was almost no region of the solution where EO was absent (Supplemental Fig. 4) as shown by the width of $EO_{40}EE_{37}$'s distribution in Supplemental Fig. 5. This behavior suggests that $EO_{40}EE_{37}$ should be equilibrated separately with a thick water layer before being combined with proteins via MCA.

Although we modeled the same CHL1/POPC/PSM membrane and proteins as Nawrocki et al.[59], we observed overall higher protein aggregation. For example, their contact fraction between villin and other villin molecules (villin-villin contacts) ranges between 25-30% whereas ours is between 27–73%. Although we both report overall higher protein contact probabilities with increasing protein concentrations, there are exceptions to this trend: Nawrocki et al. show a statistically insignificant decrease in villin-villin contact between 5% and 10% v/v. In contrast, only our protein G contact fraction decreases

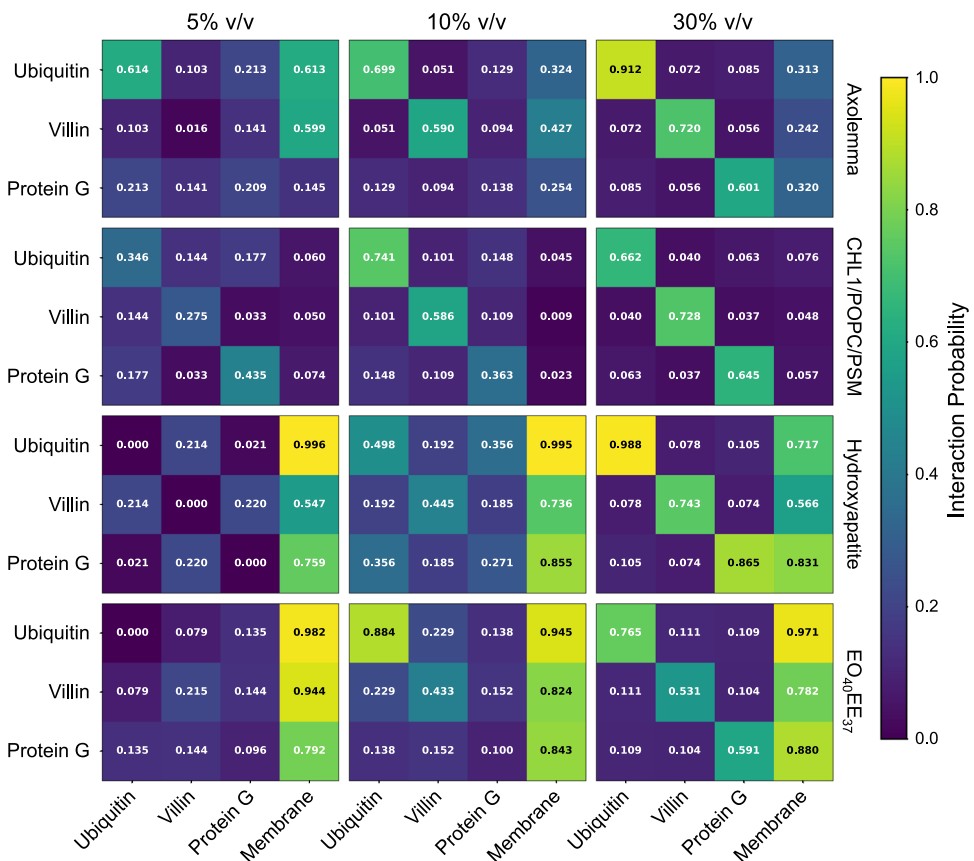

**Fig. 6 | Protein contact probability for all systems containing membrane-like components.** A contact is defined when Cα-Cα distance ≤ 7 Å. Note that for calculation of % v/v in CHL1/POPC/PSM systems only, protein volume includes the solvent accessible volume, as described in STEP 1. CHL1: cholesterol, POPC: 1-palmitoyl-2-oleoyl-phosphatidylcholine, PSM: palmitoylsphingomyelin (PSM). EO₄₀EE₃₇: polyethylene oxide-poly(ethylethylene).

between 5% and 10%. Notably, they used the CHARMM36 FF with modified ion parameters, and they also decreased protein aggregation by multiplying protein-water Lennard-Jones (LJ) potentials by 1.09, whereas we used the CHARMM36m FF that uses an updated CMAP potential to reduce left-handed α-helix formation but does not directly address protein-water interaction strength by default[47]. Due to these methodological differences, a direct comparison of our results with those of Nawrocki et al. is challenging. Further study could see if reintroducing water interaction scaling reconciles this discrepancy.

## POPC diffusion near mica-supported lipid bilayers

Planar-supported lipid bilayers (SLBs), artificial membranes consisting of a lipid bilayer deposited on a solid support, have become an essential tool for investigating the properties and functions of biological membranes. Compared to other membrane models, such as giant unilamellar vesicles or cell membranes, SLBs offer numerous advantages, including their simplicity, reproducibility, and versatility in analytical techniques, such as fluorescence microscopy, surface plasmon resonance, and atomic force microscopy[60]. Therefore, SLBs have been widely utilized in the fields of biophysics, biochemistry, and cell biology to study a variety of membrane-related processes, such as membrane protein function, protein-lipid interactions, and membrane fusion. Therefore, understanding the lateral motion of lipids in SLBs is crucial in elucidating the properties and functions of biological systems. The lateral mobility of lipids is a key determinant of membrane structure and function and is influenced by various factors such as temperature, lipid composition, and the presence of membrane proteins. Thus, accurate measurements of lipid diffusion coefficients are necessary to fully understand membrane behavior and corresponding biological events.

In this study, we investigated the effect of the solid support on lipid diffusion in SLBs using model systems (Fig. 1e). We prepared three systems with a pure POPC bilayer separated by 1, 2, and 3 nm from a mica support, and analyzed the diffusion coefficient of POPC lipids in the lower ($D_{B\text{-}1nm}$, $D_{B\text{-}2nm}$, and $D_{B\text{-}3nm}$) and upper ($D_{T\text{-}1nm}$, $D_{T\text{-}2nm}$, and $D_{T\text{-}3nm}$) leaflets using the Diffusion Coefficient Tool[61] plugin in VMD[37]. Our MD simulation results (Fig. 7) show that $D_{B\text{-}1nm}$, $D_{B\text{-}2nm}$, and $D_{B\text{-}3nm}$ are 4.8 μm²/s, 6.6 μm²/s, and 6.4 μm²/s, respectively (all errors are less than 0.01 μm²/s). Experimental values of the POPC diffusion coefficient measured by raster image correlation spectroscopy (RICS) of a lipid-like probe in a GUV bilayer are ~7 ± 3 μm²/s[62], implying that a bilayer on support behaves like free-standing lipids when the water thickness is more than 2 nm. In contrast, when the thickness of the water layer is less than 2 nm, there are strong interactions between the support and the lipids, which reduces the diffusion coefficient of the lipid in the lower leaflet of the SLB. This finding is consistent with the water thickness of dimyristoylphosphatidylcholine (DMPC) SLBs measured with the specular reflection of neutrons, which found that water thickness is in the range of 2 ~ 4 nm[63]. In addition to the lower leaflet, $D_{T\text{-}1nm}$, $D_{T\text{-}2nm}$, and $D_{T\text{-}3nm}$ show 5.9 μm²/s, 6.7 μm²/s, and 6.5 μm²/s, respectively (all errors are less than 0.01 μm²/s). These results suggest that the interaction between the mica support and the lower leaflet affects the diffusion of lipids in the upper leaflet as well. The decrease in $D_{T\text{-}1nm}$ is likely due to the coupling of the two leaflets via lipid-lipid interactions across the lipid bilayer. This finding highlights the importance of considering the behavior of both the upper and lower leaflets when studying the properties and functions of biological membranes in contact with solid supports.

In all simulations containing Mica, the bottom POPC leaflet is the one closest to Mica. The difference between the systems is the

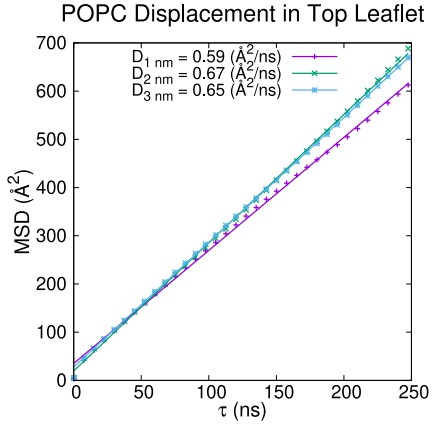
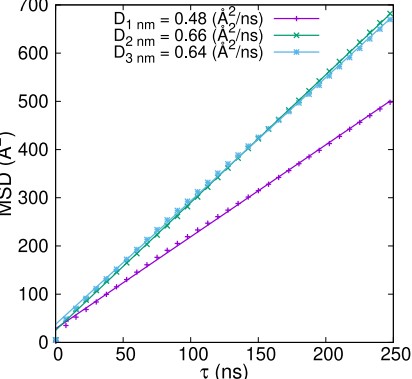

**Fig. 7 | POPC diffusion near mica-supported lipid bilayers.** Upper and lower leaflets are distal and proximal to mica, respectively. Mean square displacement (MSD) of 1-palmitoyl-2-oleoyl-phosphatidylcholine (POPC) phosphorus is plotted against lag time, with diffusion coefficient (D) shown as a fitted line. All lag times between 1 to 250 ns were used to calculate D.

thickness of the water layer separating Mica and POPC's bottom leaflet ($D_1 = 1$ nm, $D_2 = 2$ nm, $D_3 = 3$ nm). Thus, $D_1$ is the system that provides the greatest interaction between Mica and POPC. To quantify this interaction, we used the CHARMM `INTER` function to calculate interaction energies via the formula $\Delta E_{inter} = E_{both} - (E_{POPC} + E_{Mica})$ for the final frame of each simulation. CHARMM energy calculations automatically separate energy terms; the VDW ($\Delta E_{VDW}$) and electrostatic ($\Delta E_{elec}$) terms are reported in Table 2 below. We believe this interaction hinders the lateral diffusion of POPC by increasing friction between POPC and water.

### Diffusion of $CO_2$ through polymer membranes

Polyethylene terephthalate (PET) is a cheap, recyclable plastic used in containers, clothing, and many other products with the resin ID code 1. Modern industrial production and recycling of PET causes significant non-renewable energy usage (NREU) and release of greenhouse gases (GHGs)[64], which has led to an investigation into potential replacement plastics that could be constructed from renewable sources. One such candidate is poly(ethylene 2,5-furandicarboxylate) (PEF), which is amenable to production from biomass and which could substantially reduce NREU and GHG production[65]. In experimental investigations of the permeability of plastics to $CO_2$ and other gases, PEF has shown uniform improvement over PET in its $CO_2$ retention ability[66]. To investigate possible mechanisms for $CO_2$ diffusion through PET and PEF, we simulated 100 Å thick polymer sheets of each plastic with 1 atm initial pressure in pure $CO_2$ (Fig. 1f).

Our results show that $CO_2$ penetration in PEF95 is comparable to that in PET95 at each measured simulation time point. In our simulations, $CO_2$ molecules made rapid jumps between defects located at varying depths in the polymer structures, followed by long periods where the molecules remained in the same approximate region. The exact location of these defects varies greatly between replicas, as indicated by the error bars in Fig. 8, but the general trend for both polymers is an increase in overall $CO_2$ diffusion over time. Notably, between 666–1333 ns, the $CO_2$ density near the polymer center (0–12 Å) was higher for PET95 than for PEF95, and the $CO_2$ density near the polymer periphery (15–37 Å) was lower for PET95 than for PEF95 in the same time range, indicating that PEF95 is overall more resistant to $CO_2$ diffusion. Indeed, $CO_2$ diffusion we measured in the polymer center was nearly twice as high for PET95 ($7.8 \pm 2.2$ cm$^2$/s $\times 10^{-8}$) versus PEF95 ($4.0 \pm 0.3$ cm$^2$/s $\times 10^{-8}$), as described in Supplementary Methods: $CO_2$ Diffusion (Supplementary Table 1 and Supplementary Fig. 6).

### Comparison with other programs

Several existing programs facilitate densely packed macromolecular system modeling. For example, PACKMOL and cellPACK can pack molecules into complex shapes such as budding vesicles and viral capsids[10,11,24]; polyply and pysimm can model long polymer chains (e.g., DNA/RNA) densely packed with proteins in a cytoplasm[29,67]. Moltemplate can also model long polymers, but collision detection must be handled externally[68]. Supplementary Table 2 summarizes our analysis on the capabilities of each program in multicomponent assembly, which is elaborated below.

Pysimm is a Python API that facilitates modeling and simulation with LAMMPS. It can read molecular structures in several formats including PDB, XYZ, and MOL/MOL2. It can construct LAMMPS-compatible topologies and supports creating and positioning long polymers. Pysimm includes many functions that delegate common simulation tasks to LAMMPS, such as molecular dynamics and minimization. Pysimm is similar to pyCHARMM[69] embedding CHARMM functionality in a Python framework.

Although moltemplate was designed for custom CG modeling, it has also been used in the preparation of all-atom models[70]. However, external tools are required to select appropriate FF atom types and resolve collisions in prepared models. According to their website, moltemplate "is not suitable for all-atom protein simulations". When utilizing moltemplate's linear stacking method to generate initial polymer conformations, long equilibration simulations of tens to hundreds of nanoseconds are required before the melt is well relaxed. In contrast, the CHARMM-GUI Polymer Builder[30] creates initial structures similar to fully relaxed configurations, allowing for direct production runs without long equilibration simulations. As described in our recent publication, the initial polymer configurations generated by CHARMM-GUI Polymer Builder exhibit structures similar to fully relaxed configurations, allowing for direct production runs without additional equilibration simulations. However, when utilizing moltemplate's linear stacking method, additional equilibration simulations of tens to hundreds of nanoseconds are required even after the initial structure formation.

Polyply has been developed to generate input files and starting coordinates for polymeric molecules at CG and all-atom resolutions.

**Table 2 | Interaction energies between POPC and Mica in kcal mol$^{-1}$**

|  | $D_{1 \, nm}$ | $D_{2 \, nm}$ | $D_{3 \, nm}$ |
|---|---|---|---|
| $\Delta E_{total}$ | −88.36 | $-4.65 \times 10^{-3}$ | 0 |
| $\Delta E_{VDW}$ | −29.34 | $-3.95 \times 10^{-3}$ | 0 |
| $\Delta E_{elec}$ | −59.01 | $-7.08 \times 10^{-4}$ | 0 |

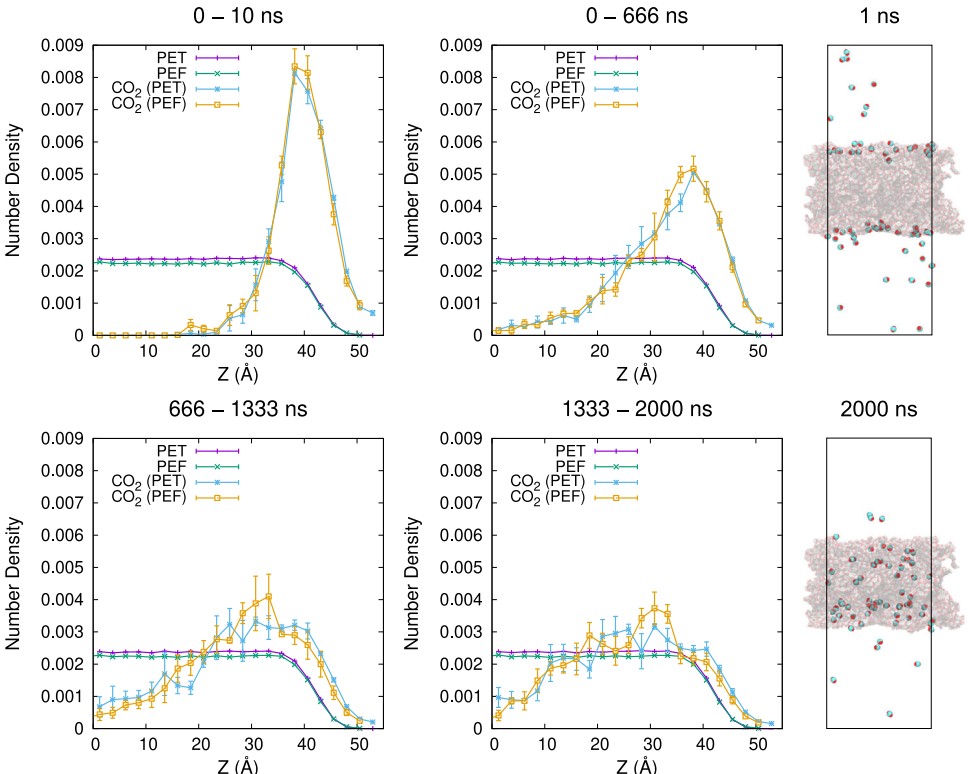

**Fig. 8 | Symmetrized Z density profiles of $CO_2$ in $PET_{95}$ and $PEF_{95}$.** Left and middle: mean ± SEM of 3 independent simulation replicas of the same plastic are plotted with a bin size of 2.46 Å. PEF ($PEF_{95}$ or polyethylene 2,5-furandicarboxylate) and $CO_2$ in $PEF_{95}$ values are shifted left by 2.46 Å to correct for changes in plastic thickness that occurred during equilibration in a vacuum. Bin heights are normalized to sum to 1. Right: simulation snapshots of PET ($PET_{95}$ or polyethylene terephthalate) and $CO_2$ taken after 1 ns and 2000 ns of NVT simulation.

However, it does not perform rigid body packing of large molecules and requires third-party software for certain biomolecules, similar to Moltemplate.

PACKMOL has been developed for finding collision-free rigid body packing solutions of arbitrary molecules in many non-periodic geometries. It implements an intuitive scripting format for describing the geometrical requirements and can read input coordinates from PDB, tinker, XYZ, and moldy formats. PACKMOL has quick runtimes with stable performance when using default settings. This makes it an ideal candidate for benchmark comparison with MCA's packing algorithm. As both MCA and PACKMOL treat molecules as rigid bodies, neither software package is well-suited to mixing long polymers with proteins.

To test whether our packing approach can lead to more dense packing and lower runtimes even though the CHARMM executable is not optimized for packing, we selected two sets of macromolecules shown in Supplementary Table 3: one where all molecules have asphericity between 0.07 – 0.15 (easy), and the other where asphericity varies between 0.02 – 0.48 (hard). We then ran 12 replicas of both macromolecule sets through PACKMOL and MCA's packing procedure with volume fraction (v/v) starting from 10% and increasing by 1% until all 12 replicas failed at the same v/v. As shown in Supplementary Fig. 1, MCA's runtimes were longer for low-density systems, and shorter for high-density systems. The maximum v/v achieved by PACKMOL was 30% (easy) and 18% (hard), and it was 41% (easy) and 23% (hard) for MCA. These results show that our approach can improve packing performance for periodic geometries, especially when the system density is high.

**Limitations**

Although there is a large library of molecules that can be handled by MCA via other CHARMM-GUI modules, incorporating molecules that are not available in CHARMM-GUI—such as new membrane lipids,

synthetic polymer building blocks, or ligands unsupported by CGenFF—could be challenging for users. Although users can download the CHARMM scripts used to generate a system, the scripts are often hundreds to thousands of lines long. While MCA allows users to upload their own CHARMM topologies (RTF) and parameters (PRM) for molecules not contained within the CHARMM36(m) or INTERFACE FFs, parameterizing molecules for the CHARMM FF is challenging. However, other programs (e.g., psfgen[71] for PSF; FFParam[12], CGenFF[16–18]) are capable of producing these topology and parameter files.

The rigid body packing used by MCA or PACKMOL tends to work well for packing problems involving macromolecules that are already fairly rigid, such as proteins, crystals, and short nucleic acid sequences. Packing long flexible molecules such as synthetic and nucleic acid polymers is unlikely to succeed because the molecule's ability to pack tightly with other components comes from its flexibility. In such cases, the polymers should be assembled by another method, such as the random walk algorithms used by pysimm[29] and polyply[28] or the CG Kuhn fragment equilibration method used by Polymer Builder[30].

## Discussion

This work presents a guided procedure for building simulation-ready, atomistic models containing heterogeneous molecular components via Multicomponent Assembler (MCA) in CHARMM-GUI. Initial positioning is facilitated by assigning component types that determine the packing strategy and using a greedy packing algorithm whose initial configuration is generated from CG simulation. The provided positioning options allow constraining components to a given position or orientation while allowing them to move within those constraints during packing.

Although MCA requires that input structures be provided in CHARMM format, other CHARMM-GUI tools—especially PDB Reader & Manipulator[56], Ligand Reader & Modeler[21], Glycan Reader & Modeler[57],

Nanomaterial Modeler[46], and Polymer Builder[30]—can be used to prepare these initial structures. Furthermore, its integration with FF-Converter[31,32] allows MCA to provide simulation inputs using both CHARMM and AMBER FFs and for many other simulation programs, including NAMD[36], GROMACS[34,35], Amber[15], and OpenMM[38]. MCA's diverse modeling capabilities are demonstrated by building crowded solute-membrane interfaces, nano-bio interfaces, multi-membrane systems, and non-water solvents. Our modeling of polymer-bio and nano-bio interfaces is made possible by the CHARMM36(m)[47] and INTERFACE FFs[50–55] and greatly facilitated by Polymer Builder[30] and Nanomaterial Modeler[46].

This work solves a PBC-aware packing problem for large, densely packed molecules. In combination with other CHARMM-GUI modules, MCA can generate complex molecular systems by combining many components. We hope that MCA can facilitate innovative studies of complex interactions between small (organic and inorganic) molecules, biomacromolecules, polymers, and nanomaterials.

## Methods

### Test system preparation
To test MCA's ability to generate complex multicomponent systems reliably, we have built and simulated the systems shown in Fig. 1. Each system's configuration is summarized in Supplementary Table 4, and the preparation is described below. Video demos of MCA usage for most of the below molecular configurations are available at https://charmm-gui.org/demo. The FFs necessary for nanomaterial and polymer modeling[50–55] are automatically included in the systems prepared by CHARMM-GUI. Note that protein-water interaction scaling previously used to reduce protein-protein interactions[72] was not used in this study.

### Solution systems with three proteins (5, 10, 30% v/v)
We built three systems described by Nawrocki et al.[59], each containing ubiquitin (PDB ID: 1UBQ), villin (PDB ID: 1VII), and streptococcal G protein (PDB ID: 3GB1) in volume fractions (v/v) of 5%, 10%, and 30%, solvated by TIP3P water and 150 mM KCl (Supplementary Table 1). Each system was simulated with OpenMM using the NVT (constant particle number, volume, and temperature) equilibration and NPT (constant particle number, pressure, and temperature) production protocols provided by CHARMM-GUI Solution Builder[31,32,45] at 310.15 K with hydrogen mass repartitioning (HMR)[73] for 1 μs. Preparation of a similar system is demonstrated in the video demo "Solvating Multiple Proteins" (https://charmm-gui.org/demo/multicomp/2). To rebuild these systems, use the corresponding volume fraction and component counts shown in Supplementary Table 4 on the STEP 1 page.

### Membrane systems with three proteins (5, 10, 30% v/v)
We built three systems containing the same proteins used in the solution systems, combined with a membrane containing an equal ratio of cholesterol (CHL1), 1-palmitoyl-2-oleoyl-phosphatidylcholine (POPC), and palmitoylsphingomyelin (PSM); see Supplementary Table 4 for system information. Membranes were built de novo in MCA using the Membrane Builder protocol in STEP 3. Systems were simulated with OpenMM using the multi-step NVT/NPT equilibration and NPT production protocols provided by Membrane Builder[40,41] at 310.15 K with HMR for 1 μs. As the CHL1/POPC/PSM membrane system was not pre-equilibrated, its size changed significantly throughout the first 100 ns of simulation. In each system, the X and Y axes shrank while the Z axis grew (Supplementary Fig. 7). Preparation of a similar system is demonstrated in the video demo "Solvating Proteins with Membranes and Membrane Proteins" (https://charmm-gui.org/demo/multicomp/3). To rebuild these systems, use the corresponding volume fraction and component counts shown in Supplementary Table 4 on the STEP 1 page, and instead of uploading 5O8F (as

shown in the demo), check the box labeled "Generate a membrane for this system" to enable the membrane size options.

### Pre-equilibrated axolemma membrane systems with three proteins (5, 10, 30 % v/v)
An axolemma membrane model that was simulated by Lee et al.[41] was prepared by manually removing water and ions from the last simulation frame. Its PSF/CRD files were then uploaded with those of 1UBQ, 1VII, and 3GB1. We used the solvated component type for the three proteins and the periodic type for the axolemma membrane. Dimensions for the axolemma were taken from the previous simulation (for X and Y) and the membrane's Z dimension was estimated as ~50 Å. Supplementary Table 4 summarizes the final system information. To speed up packing for the 5% and 10% v/v cases, protein components were excluded above and below 10 Å from the membrane region. Systems were simulated with OpenMM using the multi-step NVT/NPT equilibration and NPT production protocols provided by Membrane Builder[40,41] at 310.15 K with HMR for 1 μs. Preparation of a similar system is demonstrated in the second example from video demo "Solvating Proteins with Membrane-like Polymers or Pre-Equilibrated Membranes" (https://charmm-gui.org/demo/multicomp/4) and from video demo "Building Nano-Bio Interface with Image Bonds" (https://charmm-gui.org/demo/multicomp/5). To rebuild these systems, use the corresponding volume fraction and component counts shown in Supplementary Table 4 on the STEP 1 page.

### HAP with three proteins (5, 10, 30% v/v)
We first constructed a hydroxyapatite (HAP) slab with a dimension of 103.6 Å × 114.2 Å × 42 Å at pH 10 using Nanomaterial Modeler[46]. The HAP slab was then uploaded with the same proteins used in the solution systems. The solvated component type was used for the three proteins, and the periodic type was used for HAP. To achieve protein volume fractions of 5/10/30, the number of protein copies (2, 4, and 17 each) and system Z dimension were varied (Supplementary Table 1). To speed up packing for all cases, protein components above and below 5 Å from the HAP region were excluded. After system assembly, 5000 steps of steepest descent (SD) minimization was followed by 5000 steps of adopted basis Newton-Raphson (ABNR) minimization using CHARMM. The systems were then simulated without HMR at 303.15 K for 1 μs using OpenMM. Preparation of a similar system is demonstrated in the second example from video demo "Solvating Proteins with Membrane-like Polymers or Pre-Equilibrated Membranes" (https://charmm-gui.org/demo/multicomp/4). To rebuild these systems, use the corresponding volume fraction and component counts shown in Supplementary Table 4 on the STEP 1 page.

### Polymer $EO_{40}EE_{37}$ with three proteins (5, 10, 30% v/v)
We first constructed a polyethylene oxide-poly(ethylethylene) polymer slab ($EO_{40}EE_{37}$) in solution using Polymer Builder, as described by Choi et al.[30], with a thickness of 120 Å and a width of 107.9 Å. The slab was then uploaded with the same proteins used in the solution systems. The solvated component type and periodic type were used for the three proteins and the $EO_{40}EE_{37}$, respectively. To achieve protein volume fractions of 5/10/30, the number of protein copies (2, 4, and 13 each) and system Z dimension were varied (Supplementary Table 4). To speed up packing for the 5% and 10% v/v cases, protein components were excluded above and below 10 Å from the polymer region. After system assembly, we performed 5000 steps of SD minimization followed by 5000 steps of ABNR minimization using CHARMM. Systems were then simulated without HMR at 303.15 K for 1 μs. Preparation of a similar system is demonstrated in the second example from video demo "Solvating Proteins with Membrane-like Polymers or Pre-Equilibrated Membranes" (https://charmm-gui.org/demo/multicomp/4). To rebuild these systems, use the corresponding volume fraction and component counts shown in Supplementary Table 4 on the STEP 1 page.

## Diffusion of $CO_2$ through polymer membranes

We used Polymer Builder to construct slabs containing polyethylene terephthalate ($PET_{95}$) and polyethylene 2,5-furandicarboxylate ($PEF_{95}$) with a monomer length of 95 each in a vacuum with a thickness of 100 Å and width near 100 Å. This resulted in 40 molecules of $PET_{95}$ with a width of 108.4 Å and 46 molecules of $PEF_{95}$ with a width of 108.4 Å. We ran equilibration and 500 ns of production for each slab separately in a vacuum using the default inputs provided by Polymer Builder at 298.15 K without HMR. We obtained a $CO_2$ structure from the CHARMM36 FF by Ligand Reader & Modeler. To combine each polymer with $CO_2$, we used the solvent component type for $CO_2$ and the periodic type for $PET_{95}$ or $PEF_{95}$. Three replicas of each polymer + $CO_2$ system were constructed (Supplementary Table 4). Instead of a water solvent, we used the $CO_2$ to construct a gaseous solvent with pure $CO_2$ present at 1.98 g/L density. This resulted in 64 copies of $CO_2$ with $PET_{95}$ and 64 copies of $CO_2$ with $PEF_{95}$. After system assembly, we performed 5000 steps of SD minimization followed by 5000 steps of ABNR minimization using CHARMM. Systems were then simulated without HMR at 298.15 K for 2 µs using OpenMM. Preparation of the $PET_{95}$ example is demonstrated in the third example of the video demo "Custom Solvent Composition, Gaseous Solvents" (https://charmm-gui.org/demo/multicomp/6). Preparation of the $PEF_{95}$ example is the same except that the PEF monomer is chosen during polymer generation.

## Multi-layer system (mica + POPC membrane)

To demonstrate the capability of MCA to build multi-layer models, we obtained a mica model of 103.8 Å× 108.2 Å× 29.9 Å from Nanomaterial Modeler and uploaded it to MCA using the periodic component type and selecting the option to generate a new membrane. To analyze the effect of mica on the membrane, we generated three such models by varying the Z position of the uploaded mica's COM, $Z_1 = 48.33$ Å, $Z_2 = 58.33$ Å, and $Z_3 = 68.33$ Å, corresponding to 10 Å, 20 Å, and 30 Å initial separation between membrane head group atoms (initialized near $Z = 19$ Å) and mica. A pure POPC bilayer containing 165 lipids in each leaflet was constructed with the membrane centered at $Z = 0$. The systems were solvated with 0.15 M KCl followed by 100 steps of SD minimization in CHARMM and 5000 steps of minimization in OpenMM. Supplementary Table 4 summarizes the final system information. The mica + POPC systems were equilibrated in a multistage procedure starting with 250 ps NVT simulation at 298.15 K with positional and dihedral restraints starting at 1000 kJ/mol/nm$^2$ and 1000 kJ/mol/rad$^2$ decreasing halfway to 400 kJ/mol/nm$^2$ and 400 kJ/mol/rad$^2$. We then used an NPT ensemble at 298.15 K for 3.625 ns with restraints decreasing from 400 kJ/mol/nm2 and 200 kJ/mol/rad2 to 0 and pressure coupling applied in all dimensions at 1 atm, as shown in Supplementary Fig. 8. Finally, all systems were run for 1 µs under NVT ensemble at 298.15 K without HMR. The exact steps required to generate these examples in MCA are described in Supplementary Methods: Mica + POPC Generation.

## MCA and PACKMOL packing benchmark environment

**Computing environment.** We used one Dell XPS 8900 workstation with an Intel Core i7-6700 CPU (4 cores) and 8 GB of installed RAM. On the Ubuntu 22.04.2 LTS operating system, we installed PACKMOL 20.14.2 and CHARMM 48a1 (commit ID cb36cf5c6) from CHARMM's development branch.

To set an appropriate number of simultaneous tests to run on the workstation, we observed CPU and memory usage of solo tests and determined that while PACKMOL and CHARMM consistently consumed 1 CPU at a rate near 100%, PACKMOL's memory usage was uniformly below 9 MB, whereas CHARMM tests used 3.5 GB. To make the most of the workstation's resources, we thus ran 4 simultaneous jobs when testing PACKMOL and 2 when testing CHARMM.

In all tests, target system densities were achieved by varying total system volume while keeping the number of molecules constant. All code used to run our benchmark can be found at https://github.com/charmm_gui/mca_scripts[74].

**MCA.** One MCA job was created on CHARMM-GUI for each set of macromolecules shown in Supplementary Table 3. After selecting the number of molecules and a 10% volume fraction, we performed packing in the web GUI and downloaded the projects from the solvent options page without generating a solvent. Each macromolecule set's project directory was copied to create 12 replicas (24 total). The directories were copied again to create one directory for each tested volume fraction at 1% intervals starting from 10% and increasing until all replicas failed at the same v/v. Random number generator (RNG) seeds were set automatically from CPU time.

**PACKMOL.** To avoid giving MCA an undue advantage, hydrogen atoms were stripped from PACKMOL's input PDB files, as MCA ignores hydrogen atoms when checking for collisions. Similarly, the collision tolerance was set to 2.5 Å to match MCA's tolerance. RNG seeds were set automatically from CPU time. Default values were used for all other settings. As with MCA, 12 replicas of each macromolecule set were tested at 1% v/v intervals starting from 10% and increasing until all replicas failed at the same v/v.

## Reporting summary

Further information on research design is available in the Nature Portfolio Reporting Summary linked to this article.

## Data availability

Initial simulation structures, simulation parameters, and programs used to produce figures are included in the GitHub repository described in Code Availability. Any additional data are available upon request. PDB structures referenced in this work: 1MJC, 1UBQ, 1VII, 2HAC, 3GB1, 6Y3G. A source data archive (source_data.tgz) was added to the github repository at [https://github.com/charmmgui/mca_scripts/raw/main/source_data.tgz]. Version of code submitted for publication can be found in the Zenodo repository [10.5281/zenodo.11205908].

## Code availability

CHARMM-GUI's full licensing terms are shown here [https://charmm-gui.org/?doc=license_cgui]. CHARMM-GUI is free for academic and governmental use, but not for commercial use or any collaboration with commercial entities. These terms apply to the usage of our web service CHARMM-GUI and to any output files we provide (e.g., CHARMM scripts). Scripts used to manipulate PSF/CRD files before upload to MCA are available at [https://github.com/charmm-gui/mca_scripts][74].

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

## Acknowledgements

This work has been supported by NIH GM138472 and NSF OAC-1931343.

## Author contributions

N.K., Y.K.C., J.L., and W.I. designed the research. N.K. performed the research, wrote the software, and drafted the paper with input from the co-authors. Y.K.C. wrote the text for POPC diffusion near mica-supported lipid bilayers. J.L. helped with FF Converter integration. Y.K.C. and J.L. recommended appropriate default settings for simulation inputs. Y.K.C., J.L., and W.I. helped with the interpretation of results and substantially revised the manuscript.

## Competing interests

W.I. is the co-founder and CEO of MolCube INC. J.L. is the co-founder and CTO of MolCube INC. The remaining authors declare no competing interests.
