## [Peer Review File · Nature Communications]

CHARMM-GUI Multicomponent Assembler for Modeling and Simulation of Complex Multicomponent SystemsREVIEWER COMMENTS

Reviewer #1 (Remarks to the Author):

This is a valuable contribution from the developer of the immensely popular and successful computational tool, CHARMM-GUI, that describes the implementation of the sophisticated methodologies for the assembly of multi-component (bio)molecular systems for simulation studies. The significance of such work can not be overstated, since it not only provides a powerful tool for researchers to prepare complex molecular simulations, but also ensures the reproducibility of such simulations. The latter is an issue of major concern for the computational analysis of highly complex (bio)molecular systems. The versatility of the approach is well demonstrated by the analysis of a range of complex systems involving different proteins, lipids, polymers and solid. From a technical point of view, the study overcomes the challenge of conducting sophisticated packing calculations under periodic boundary condition. The efficiency and robustness of the approach make the tool of great value to the broad computational chemistry/materials community.

Overall, the methodology is clearly described and the examples are clearly discussed. The amount of novel mechanistic insights from these examples is limited, but this is not the focus on the study, which reports an enabling methodology. Considering the significant impact of the tool on the diverse computational community, I support publication of the work in a high profile place such as Nature Communications.

Reviewer #2 (Remarks to the Author):

In this paper, Kern et al. present and demonstrate the algorithm, which underlies the CHARMM-GUI Multicomponent Assembler (MCA). The MCA is a webserver-based workflow for assembling coordinates and generating simulation input parameters necessary to start molecular dynamics simulations of complex systems. The MCA workflow is the latest among a long list of CHARMM-GUI tools and specifically implements the following new features: (1) a PBC-aware packing algorithm; (2) a generalized template-based solvent approach for arbitrary mixtures of solvent molecules; (3) a PBC-aware utilization of a periodic membrane.

A program or workflow for packing multicomponent systems is in itself not novel and many tools have been presented in the literature. The introduction gives an adequate overview of the most popular tools in the field of biomolecular simulations. A minor point of critique is the lack of tools from the material science side – especially for generating systems containing polymers, which is one of the test cases presented later. For example, moltemplate1, polyply2, pysimm3, are all programs that could be used to create example case 4.

The authors identify several shortcomings of the existing tools that they aim to resolve with the MCA. Those shortcomings are the lack of PBC-aware packing, “significant experimental knowledge, manual intervention, or use of ad hoc scripts” to be able to generate starting coordinates and simulation parameters, as well as the need for “significant post-processing”. The lack of PBC-aware packing and the fact that potentially more than one program/script has to be used to generate both simulation input parameters and coordinates, I consider valid shortcomings of the existing tools mentioned. However, this paragraph contains some inappropriate generalizations. For instance, what constitutes “significant experimental knowledge” that the MCA does not require? What is “manual intervention” in the context of generating coordinates and topologies? Doesn't the MCA also require the user to generate the appropriate CHARMM input in multiple manual steps? To consider the paper for publication I strongly recommend working out more clearly what MCA brings to the table and rephrasing the previously mentioned generalizations or supporting them by references to literature. Considering the widespread use of CHARMM and the CHARMM-GUI, it is of significant interest to the scientific community to have a tool such as MCA, which allows the combination of the different CHARMM-GUI modules in a consistent manner. Similar papers describing tools to set up MD simulations have already been published in the journal at hand.^{2,4} However, it needs to be kept in

mind that the new features the MCA offers are the assembly and packing of the different components. Protocols for generating proteins, membranes, or polymers have already been published previously.^{5,6} At the same time there are several shortcomings of the current manuscript, which in my opinion need to be improved before the paper can be published:

(1) A core improvement of MCA over existing tools is the PBC-aware packing algorithm. While the six benchmark cases are complex enough to support the claim of wide-spread applicability, the manuscript lacks proper benchmarks for the packing algorithm itself. For the highly concentrated protein systems, the authors choose three rather small and spherical proteins to pack. I'd argue that these proteins are therefore quite easy to pack. A proper benchmark should consider a wider set of diversly shaped proteins and assess the speed comparison to existing tools (take Packmol7 for example), and the resilience (i.e., how often does it crash). A diverse set of proteins that is biologically relevant as well are the cellular proteins of the Syn3A minimal cell. Interestingly, the protein content of this cell is also about 30% of the volume ⁸, which is the upper limit for efficient packing using MCA. I also suggest reporting the sphericity or asphericity of the proteins.

(2) Along the lines of the previous point, the authors mention in their demo video that the packing algorithm can fail if solution components are too close to the membrane or "bad" lipids are selected. A proper benchmark of how robust the packing algorithm is considering more difficult lipid membranes (e.g., containing largish glycolipids and larger transmembrane proteins) seems appropriate to substantiate their claim that MCA does not require significant manual intervention. Repeated trial and error for generating coordinates to me is not so different from having to use multiple scripts and can be rather daunting for non-expert users. This benchmark appears especially important as non-expert users appear to be one of the target groups for MCA.

(3) While it seems evident that the MCA can generate combinations of pre-build polymer systems and proteins, I wonder if it is possible to generate a protein solution in the presence of polymers. For example, a protein solution with crowders such as PEG is a frequently used cell mimetic. Can the MCA build such a system? Other tools such as moltemplate or polyply could for example be used for such systems.

(4) The complete lack of a discussion section, which also provides some limitations of the MCA, surprises me. I strongly recommend adding a discussion that sets into perspective the advantages of MCA in contrast to its limitations. In relation to existing tools, there are several limitations of the MCA in my opinion: Currently, it is not possible to easily implement molecules unknown to the CHARMM-GUI such as new lipids or polymers. While other tools may require some more knowledge, they are generally extendable, and implementing new molecules is well-documented. Furthermore, the support of an API makes it impossible to use MCA for automated pipelines or high-throughput workflows, which are more easily accessible in the other programs. For membranes and solutions, there exists a high throughput simulator. Can the MCA be connected to it in the future?

(5) MCA when used together with other tools such as membrane builder can take several hours up to days in order to generate a morphology. To my knowledge, this performance is slower than that of comparable tools such as polyply for polymers or cellPack (especially with GPU acceleration) for molecular solutions. Of course, the advantages of MCA might well outweigh the disadvantages but this bit deserves a discussion together with the limitations in point 4.

(6) Finally, in their reporting summary, the authors give an MIT license for their code. I'm confused as to what this means. Is the CHARMM-GUI MIT licensed or is the code openly available under MIT license? It is important to state clearly under which conditions the code is available because to my knowledge the actual code is not publicly available. If it is, I'd recommend mentioning it more clearly in the manuscript.

References

1. Jewett, A. I. et al. Moltemplate: A Tool for Coarse-Grained Modeling of Complex Biological Matter and Soft Condensed Matter Physics. *J Mol Biol* 433, 166841 (2021).
2. Grünwald, F. et al. Polyply; a python suite for facilitating simulations of macromolecules and nanomaterials. *Nat Commun* 13, 68 (2022).
3. Fortunato, M. E. & Colina, C. M. pysimm: A python package for simulation of molecular systems. *SoftwareX* 6, 7–12 (2017).
4. Pezeshkian, W., König, M., Wassenaar, T. A. & Marrink, S. J. Backmapping triangulated surfaces to

- coarse-grained membrane models. *Nat Commun* 11, 1–9 (2020).
- Choi, Y. K. et al. CHARMM-GUI Polymer Builder for Modeling and Simulation of Synthetic Polymers. *J Chem Theory Comput* 17, 2431–2443 (2021).
 - Lee, J. et al. CHARMM-GUI Input Generator for NAMD, GROMACS, AMBER, OpenMM, and CHARMM/OpenMM Simulations Using the CHARMM36 Additive Force Field. *J Chem Theory Comput* 12, 405–413 (2016).
 - Martínez, L., Andrade, R., Birgin, E. G. & Martínez, J. M. PACKMOL: a package for building initial configurations for molecular dynamics simulations. *J. Comput. Chem.* 30, 2157–2164 (2009).
 - Stevens, J. A. et al. Molecular dynamics simulation of an entire cell. *Front Chem* 11, (2023).

Reviewer #3 (Remarks to the Author):

In this manuscript Kern et al. introduce a novel tool in CHARMM-GUI to enable the assembly of heterogeneous multicomponent systems and input preparation using PBC, which they validate using six model systems. I consider this a step forward in the applicability of the platform thereby enhancing the versatility of the CHARMM-GUI. However, there are some points that in my view require the attention of the authors.

- The nomenclature is not well defined, e.g. NP in the introduction, EO40EE37, NPAT, MEMB etc. The authors should either define all terms beforehand or introduce a table with the definitions in the manuscript.
- The authors mention that “no free software package can handle all steps of model preparation without significant experimental knowledge, manual intervention, or use of ad hoc scripts” and underline at several locations that they aim to lower the entrance barrier for newbies in the field. I consider that the tone of these affirmations should be tuned down a bit. While I agree with the authors that at times a lower barrier might be useful, the user must have fundamental knowledge of their system, what they want to achieve and what the individual methods employed do. Hence, the user should not be encouraged to use the proposed package as a black box, which is what the strength of these sentences underline.
- Please cite the appropriate papers for all the mentioned software, e.g. PACKMOL, cellPACK, LipidWrapper etc.
- Please annotate/label the steps to be followed also in Fig. 1.
- Given that the input still requires quite some manipulation of the files, it would be useful to include a concrete example, including scripts, clear commands/snapshots in the SI.
- Typo “polling”
- At times more explanation is needed, e.g. “first letter corresponds to the segment type (P, D, R, H, C, W)” – what do these letters concretely represent? It would be useful to double check and clarify the explanations of Step 1.
- Step 2: It is unclear how the *sasa* is calculated. Is this intrinsically implemented or are there additional steps the user needs to assess? Also, why is achieving a volume fraction higher than 30% a trial-and-error process and how can this be avoided?
- Before figure 5 the authors mention that the CG representation is replaced with all-atom models. How is this concretely done? Additional details are required.
- Protein-protein and protein-membrane contacts
 - As the authors choose to reproduce already published results as validation, they should compare their results to those in Ref. 59.
 - I am curious why in Fig 6 for the second row the probability for Ubiquitin-membrane interaction decreases at 10% and then increases at 30% (comparable to 0%). Since the last paragraph is very descriptive, I would appreciate an interpretation of the results also in light (and added value) to Ref. 59.
- All the simulation details for all the different systems the authors use as validation should be described, including force-fields, simulation length, water models, ions etc. This is very important for the reproducibility study as the authors directly aim to compare to something.

12. Figure 7: it is unclear why the displacement in the bottom leaflet (D1) is slower. The authors also mention that the results are comparable to experiments but there is no mention of what type of experiments these are.

13. Diffusion of CO₂

a. More details on the system and comparison to existing data (simulation, experiment?) is required. The results are descriptive and lacking interpretation, hence it is unclear why these results are relevant and/or reproduce correctly the relevant data.

14. Conclusions

a. The authors stress that the study presents an "automated and versatile procedure for building". I recommend softening these types of statements as quite some manipulation is required, a fact also acknowledged by the authors in the next paragraph.

15. General comments:

a. The figures would benefit from larger labels, thicker lines and a better choice of colors. Generally on a printed version of the manuscript most lines are not visible or indistinguishable, the labels and legends are too small.

b. In Fig. S4 It would be useful to mention if these are representative structures and if so how they were chosen. If clustering was performed then how.

Reviewer 1

We thank the Reviewer for the positive comments. Since there are no suggestions to revise the manuscript, we have no specific responses.

Reviewer 2

We thank the Reviewer for the positive comments and suggestions. We revised the manuscript according to the Reviewer's comments as much as we could, and our specific responses are given below.

Comments

In this paper, Kern et al. present and demonstrate the algorithm, which underlies the CHARMM-GUI Multicomponent Assembler (MCA). The MCA is a webserver-based workflow for assembling coordinates and generating simulation input parameters necessary to start molecular dynamics simulations of complex systems. The MCA workflow is the latest among a long list of CHARMM-GUI tools and specifically implements the following new features: (1) a PBC-aware packing algorithm; (2) a generalized template-based solvent approach for arbitrary mixtures of solvent molecules; (3) a PBC-aware utilization of a periodic membrane.

[Comment 1] A program or workflow for packing multicomponent systems is in itself not novel and many tools have been presented in the literature. The introduction gives an adequate overview of the most popular tools in the field of biomolecular simulations. A minor point of critique is the lack of tools from the material science side – especially for generating systems containing polymers, which is one of the test cases presented later. For example, moltemplate, polyply, pysimm, are all programs that could be used to create example case 4.

[Response 1] While it may indeed be the case that our example case 4 (polymer slab + proteins in separate solvated region) can be created by moltemplate, polyply, and pysimm, these programs and their documentation or example codes do not appear to contain a workflow designed to facilitate such a system. Rather, they use a more general *scripting* approach that relies on the user to design the workflow with the functions provided by each program.

In a similar sense, CHARMM—which implements a general modeling language—can be used to build the example case 4, but it required years of development for us to design the MCA workflow using CHARMM that can handle all of the example cases we presented. Note that the result of this workflow is the main content of our manuscript. If we were to include moltemplate, polyply, or pysimm in a benchmark, we would first need to design a workflow for each program. Besides the considerable efforts this would require, it is unlikely we would be able to present a fair comparison with MCA, since we are not experts in using those programs for workflow design.

For example, the work of J. A. Stevens, *et al.* (2023), referenced by the reviewer, uses experimentally obtained component positions to construct a whole cell model that is assembled with Polyply, Martinize2, TS2CG, and Bentopy (an unpublished package). While this construction is impressive and we await Bentopy's publication with anticipation, a direct comparison with MCA appears to be challenging.

As the Reviewer suggested, however, we performed the benchmark study shown below for MCA and PACKMOL using biomolecules with varying asphericity. The Syn3A model includes long chromosomes that cannot be modeled well by either MCA or PACKMOL because both programs treat molecules as rigid bodies, as explained in **Response 7**. Therefore, we chose to instead model a smaller set of representative molecules and added the following new Section 5 in Results & Discussion.

<Added>

5. Comparison with other programs

Several existing programs facilitate densely packed macromolecular system modeling. For example, PACKMOL and cellPACK can pack molecules into complex shapes such as budding vesicles and viral capsids^{10,11,24}; polyply and pysimm can model long polymer chains (e.g., DNA/RNA) densely packed with proteins in a cytoplasm^{26,29}. Moltemplate can also model long polymers, but collision detection must be handled externally⁷⁰. **Table S2** summarizes our own analysis on the capabilities of each program in multicomponent assembly, which is elaborated below.

Pysimm is a Python API that facilitates modeling and simulation with LAMMPS. It can read molecular structures in several formats including PDB, XYZ, and MOL/MOL2. It can construct LAMMPS-compatible topologies and supports creating and positioning long polymers. Pysimm includes many functions that delegate common simulation tasks to LAMMPS, such as molecular dynamics and minimization. Pysimm is similar to pyCHARMM⁷¹ embedding CHARMM functionality in a python framework.

Moltemplate is designed for CG simulations, so that third-party molecular builder tools are required to construct all-atom models of biomolecules such as proteins, RNA, and DNA. As described in our recent publication³⁰, the initial polymer configurations generated by CHARMM-GUI *Polymer Builder* exhibit structures similar to fully relaxed configurations, allowing for direct production runs without additional equilibration simulations. However, when utilizing moltemplate's linear stacking method, additional equilibration simulation of tens to hundreds of nanoseconds are required even after the initial structure formation.

Polyply has been developed for CG simulations using the Martini force field and recently extended to support all-atom models for certain long polymers and carbohydrates. However, similar to Moltemplate, third-party software is required for biomolecules.

PACKMOL has been developed for finding collision-free rigid body packing solutions of arbitrary molecules in many non-periodic geometries. It implements an intuitive scripting format for describing the geometrical requirements and can read input coordinates from PDB, tinkers, xyz, and moldy formats. PACKMOL has quick runtimes with stable performance when using default settings. This makes it an ideal candidate for benchmark comparison with MCA. As both MCA and PACKMOL treat molecules as rigid bodies, neither software package is well-suited to mixing long polymers with proteins.

To test whether our packing approach can lead to more dense packing and lower runtimes despite the fact that the CHARMM executable is not optimized for packing, we selected two sets of macromolecules shown in **Table S3**: one where all molecules have asphericity between 0.07 – 0.15 (easy) and the other where asphericity varies between 0.02 – 0.48 (hard). We then ran 12 replicas of both sets through PACKMOL and MCA with volume fraction (v/v) starting from 10% and increasing by 1% until all 12 replicas failed at the same v/v. As shown in **Figure S6**, MCA's runtimes were longer for low density systems, and shorter for high density systems. The maximum v/v achieved by PACKMOL was 30% (easy) and 18% (hard), and it was 41% (easy) and 23% (hard) for MCA. These results show that our approach can improve packing performance for periodic geometries when the system density is high.

The following elaboration of our benchmark environment was added to Methods.

<Added>

MCA and PACKMOL Packing Benchmark Environment

Computing environment

We used one Dell XPS 8900 workstation with an Intel Core i7-6700 CPU (4 cores) and 8 GB of installed RAM. On the Ubuntu 22.04.2 LTS operating system, we installed PACKMOL 20.14.2 and CHARMM 48a1 (commit ID cb36cf5c6) from CHARMM's development branch.

To set an appropriate number of simultaneous tests to run on the workstation, we observed CPU and memory usage of solo tests and determined that while PACKMOL and CHARMM consistently consumed 1 CPU at a rate near 100%, PACKMOL's memory usage was uniformly below 9 MB, whereas CHARMM tests used 3.5 GB. To make the most of the workstation's resources, we thus ran 4 simultaneous jobs when testing PACKMOL and 2 when testing CHARMM.

In all tests, target system densities were achieved by varying total system volume while keeping the number of molecules constant. All code used to run our benchmark can be found at https://github.com/charmm_gui/mca_scripts.

MCA

One MCA job was created on CHARMM-GUI for each set of macromolecules shown in **Table S3**. After selecting the number of molecules and a 10% volume fraction, we performed packing in the web GUI and downloaded the projects from the solvent options page without generating a solvent. Each macromolecule set's project directory was copied to create 12 replicas (24 total). The directories were copied again to create one directory for each tested volume fraction at 1% intervals starting from 10% and increasing until all replicas failed at the same v/v. Random number generator (RNG) seeds were set automatically from CPU time.

PACKMOL

To avoid giving MCA an undue advantage, hydrogen atoms were stripped from PACKMOL's input PDB files, as MCA ignores hydrogen atoms when checking for collisions. Similarly, the collision tolerance was set to 2.5 Å to match MCA's tolerance. RNG seeds were set automatically from CPU time. Default values were used for all other settings. As with MCA, 12 replicas of each macromolecule set were tested at 1% v/v intervals starting from 10% and increasing until all replicas failed at the same v/v.

The following figure and tables were added to Supplemental Information.

<Added>

Table S2. Analysis of various programs for multicomponent molecular assemblies. "Special script" refers to a scripting language created specifically for a given modeling program. Topology preparation is "automatic" if individual molecule topologies can be inferred or read from a database and "manual" if they must be provided separately by the user.

	MCA	Pysimm	Moltemplate	PACKMOL	Polyply
User Interface	GUI or CHARMM script	Python API	Special script	Special script	Special script
Supported Simulation Programs	GROMACS CHARMM OpenMM Amber GENESIS Desmond	LAMMPS CASSANDRA	LAMMPS	N/A	GROMACS

Supported Force Fields	CHARMM AMBER	CHARMM AMBER GAFF DREIDING PCFF	OPLS COMPASS MARTINI GAFF DREIDING	N/A	MARTINI
Topology Preparation	Automatic	Automatic	Manual	N/A	Manual

Table S3. Molecules used in benchmark. 10 copies (easy) and 8 copies (hard) of each molecule were packed into cubic geometries.

Test Name	PDB ID	Asphericity	# Residues	Volume (\AA^3)
Easy	1ubq	0.07	76	9.37×10^3
	1vii	0.14	36	4.45×10^3
	3gb1	0.15	56	6.62×10^3
Hard	1mjc	0.02	69	7.84×10^3
	3gb1	0.14	56	6.62×10^3
	1vii	0.15	36	4.45×10^3
	6y3g	0.41	87	2.15×10^4
	2hac	0.48	60	7.78×10^3

Figure S6. Comparison of performance between PACKMOL and MCA. (A) Mean \pm standard error runtime of packing tasks at a given volume fraction (% v/v). The last point in each line is the v/v at which all packing attempts failed. (B) The fraction of packing attempts that fail. Only the lowest v/v resulting in 100% failure is shown for each combination of program and molecule set.

[Comment 2] The authors identify several shortcomings of the existing tools that they aim to resolve with the MCA. Those shortcomings are the lack of PBC-aware packing, “significant

experimental knowledge, manual intervention, or use of ad hoc scripts” to be able to generate starting coordinates and simulation parameters, as well as the need for “significant post-processing”. The lack of PBC-aware packing and the fact that potentially more than one program/script has to be used to generate both simulation input parameters and coordinates, I consider valid shortcomings of the existing tools mentioned. However, this paragraph contains some inappropriate generalizations. For instance, what constitutes “significant experimental knowledge” that the MCA does not require? What is “manual intervention” in the context of generating coordinates and topologies? Doesn’t the MCA also require the user to generate the appropriate CHARMM input in multiple manual steps? To consider the paper for publication I strongly recommend working out more clearly what MCA brings to the table and rephrasing the previously mentioned generalizations or supporting them by references to literature.

[Response 2] To avoid ambiguity and overstating our case, we have revised the following sentence from Introduction.

<Before>

However, except for a few notable model types, no free software package can handle all steps of model preparation without significant experimental knowledge, manual intervention, or use of *ad hoc* scripts.

<After>

Although all software packages require familiarity with fundamentals of molecular modeling, there are still opportunities to lower the entry barrier for modeling systems with multiple proteins and/or metabolites of interest at the all-atom resolution.

[Comment 3] Considering the widespread use of CHARMM and the CHARMM-GUI, it is of significant interest to the scientific community to have a tool such as MCA, which allows the combination of the different CHARMM-GUI modules in a consistent manner. Similar papers describing tools to set up MD simulations have already been published in the journal at hand.[2,4] However, it needs to be kept in mind that the new features the MCA offers are the assembly and packing of the different components. Protocols for generating proteins, membranes, or polymers have already been published previously.[5,6]

[Response 3] MCA indeed does not re-implement the procedures in *Polymer Builder* or *Input Generator* (now *FF-Converter*), but instead builds on and integrates with them.

We have added citations for TS2CG, polyply, pysimm, and Polymer Builder to Introduction.

<Before>

Many software applications have been developed to facilitate various steps of atomistic model preparation, including FFParm¹², FFTK¹³, SwissParam¹⁴, Antechamber¹⁵, CGenFF^{16–18}, MATCH¹⁹, OpenFF²⁰, and CHARMM-GUI *Ligand Reader & Modeler*²¹ for FF preparation; PACKMOL^{22–24}, cellPACK^{10,11}, LipidWrapper²⁵, and Soup²⁶ for molecular packing; and *FF-Converter*^{27,28} and ParmEd²⁹ for preparing and converting inputs for several simulation programs.

<After>

Many software applications have been developed to facilitate various steps of atomistic model preparation, including FFParm¹², FFTK¹³, SwissParam¹⁴, Antechamber¹⁵, CGenFF^{16–18}, MATCH¹⁹, OpenFF²⁰, and CHARMM-GUI *Ligand Reader & Modeler*²¹ for FF preparation; PACKMOL^{22–24}, cellPACK^{10,11}, TS2CG²⁵, LipidWrapper²⁶, and Soup²⁷ for molecular packing; polyply²⁸, pysimm²⁹, and *Polymer Builder*³⁰ for building and assembling

long polymer chains; and *FF-Converter*^{31,32} and ParmEd³³ for preparing and converting inputs for several simulation programs.

[Comment 4] At the same time there are several shortcomings of the current manuscript, which in my opinion need to be improved before the paper can be published:

A core improvement of MCA over existing tools is the PBC-aware packing algorithm. While the six benchmark cases are complex enough to support the claim of wide-spread applicability, the manuscript lacks proper benchmarks for the packing algorithm itself. For the highly concentrated protein systems, the authors choose three rather small and spherical proteins to pack. I'd argue that these proteins are therefore quite easy to pack. A proper benchmark should consider a wider set of diversly shaped proteins and assess the speed comparison to existing tools (take Packmol[7] for example), and the resilience (i.e., how often does it crash). A diverse set of proteins that is biologically relevant as well are the cellular proteins of the Syn3A minimal cell. Interestingly, the protein content of this cell is also about 30% of the volume [8], which is the upper limit for efficient packing using MCA. I also suggest reporting the sphericity or asphericity of the proteins.

[Response 4] As the Reviewer suggested, we performed the benchmark study shown in **Response 1** using MCA and PACKMOL using biomolecules with varying asphericity. The Syn3A model includes long chromosomes that cannot be modeled well by either MCA or PACKMOL. Therefore, we chose to instead model two smaller sets of representative molecules, as explained in **Response 1**.

[Comment 5] Along the lines of the previous point, the authors mention in their demo video that the packing algorithm can fail if solution components are too close to the membrane or "bad" lipids are selected. A proper benchmark of how robust the packing algorithm is considering more difficult lipid membranes (e.g., containing largish glycolipids and larger transmembrane proteins) seems appropriate to substantiate their claim that MCA does not require significant manual intervention. Repeated trial and error for generating coordinates to me is not so different from having to use multiple scripts and can be rather daunting for non-expert users. This benchmark appears especially important as non-expert users appear to be one of the target groups for MCA.

[Response 5] We agree with the Reviewer that the level of knowledge about one's modeling goals and possible trial and error necessary to reconcile collisions between long glycolipids and solvated proteins can count as "manual intervention". Therefore, we rephrased our claims more modestly, as explained in **Response 2**.

[Comment 6] While it seems evident that the MCA can generate combinations of pre-build polymer systems and proteins, I wonder if it is possible to generate a protein solution in the presence of polymers. For example, a protein solution with crowders such as PEG is a frequently used cell mimetic. Can the MCA build such a system? Other tools such as moltemplate or polyply could for example be used for such systems.

[Response 6] MCA itself is ill-suited to create this type of system, or even the polymer slab we used in the example case 4. Our polymer was created separately by *Polymer Builder*, then uploaded to MCA. To discuss this limitation, we included the following paragraph in the new Section 6 in Results & Discussion, as explained more in **Response 7** (reprinted below).

<Added>

The rigid body packing used by MCA or PACKMOL tends to work well for packing problems involving macromolecules that are already fairly rigid, such as proteins, crystals, and short nucleic acid sequences. Packing long flexible molecules such as synthetic and nucleic acid polymers is unlikely to succeed because the molecule's ability to pack tightly with other components comes from its flexibility. In such cases, the polymers should be assembled by another method, such as the random walk algorithms used by *pysimm* and *polyply* or the CG Kuhn fragment equilibration method used by *Polymer Builder*³⁰.

[Comment 7] The complete lack of a discussion section, which also provides some limitations of the MCA, surprises me. I strongly recommend adding a discussion that sets into perspective the advantages of MCA in contrast to its limitations. In relation to existing tools, there are several limitations of the MCA in my opinion: Currently, it is not possible to easily implement molecules unknown to the CHARMM-GUI such as new lipids or polymers. While other tools may require some more knowledge, they are generally extendable, and implementing new molecules is well-documented. Furthermore, the support of an API makes it impossible to use MCA for automated pipelines or high-throughput workflows, which are more easily accessible in the other programs. For membranes and solutions, there exists a high throughput simulator. Can the MCA be connected to it in the future?

[Response 7] We eventually plan to create a web API to facilitate automated access to CHARMM-GUI. However, it will be a big project because all CHARMM-GUI modules would need to be modified to support this type of access. We do not currently have any plans to connect MCA to *High Throughput Simulator*, but we will consider the idea if we get the necessary funding and talent.

We agree that the current CHARMM-GUI workflow does not easily handle new lipid and polymer types, although CHARMM-GUI supports a diverse set of lipids and polymer monomers. MCA facilitates combining results of other CHARMM-GUI modules by accepting CHARMM-formatted topology, parameter, structure, and coordinate files produced by those modules. However, MCA does not require the files to be produced by CHARMM-GUI, only that they need to be in CHARMM format. Many other programs (e.g., *psfgen* for PSF; *FFParam*, *CGenFF*, etc. for topologies and parameters; *MDAnalysis* etc. for CRD) are capable of producing these files. Regardless, users typically must be familiar enough with CHARMM file formats to resolve file reading errors.

In particular, to address this comment, we added the following new Section 6 in Results & Discussion.

<Added>

6. Limitations

Although there is a large library of molecules that can be handled by MCA via other CHARMM-GUI modules, incorporating molecules that are not available in CHARMM-GUI—such as new membrane lipids, synthetic polymer building blocks, or ligands unsupported by *CGenFF*—could be extremely challenging for users. Although users can download the CHARMM scripts used to generate a system, the scripts are often hundreds to thousands of lines long. While MCA allows users to upload their own CHARMM topologies (RTF) and parameters (PRM) for molecules not contained within the CHARMM36(m) or INTERFACE FFs, parameterizing molecules for the CHARMM FF is challenging, but many other programs (e.g., *psfgen* for PSF⁷²; *FFParam*¹², *CGenFF*¹⁶⁻¹⁸, etc. for topologies and parameters; *MDAnalysis*⁷³ etc. for CRD) are capable of producing these files.

The rigid body packing used by MCA or PACKMOL tends to work well for packing problems involving macromolecules that are already fairly rigid, such as proteins, crystals, and short nucleic acid sequences. Packing long flexible molecules such as synthetic and nucleic acid polymers is unlikely to succeed because the molecule's ability to pack tightly with other components comes from its flexibility. In such cases, the polymers should be assembled by another method, such as the random walk algorithms used by pysimm and polyply or the CG Kuhn fragment equilibration method used by *Polymer Builder*³⁰.

[Comment 8] MCA when used together with other tools such as membrane builder can take several hours up to days in order to generate a morphology. To my knowledge, this performance is slower than that of comparable tools such as polyply for polymers or cellPack (especially with GPU acceleration) for molecular solutions. Of course, the advantages of MCA might well outweigh the disadvantages, but this bit deserves a discussion together with the limitations in point 4.

[Response 8] We agree that MCA and other CHARMM-GUI tools often have long runtimes for complex systems. However, we hope that our **Response 1** is sufficient to address this comment, in particular our inclusion of runtime results and discussion. We believe that CHARMM-GUI users are using CHARMM-GUI with some long runtimes as it is not easy to use other programs to accomplish a similar level of complexity without customized complex scripting. However, users can submit multiple jobs and return to them at their leisure through the use of Job Retriever—if they know their job's ID—or with the recently added “Job IDs” page of their user profile.

[Comment 9] Finally, in their reporting summary, the authors give an MIT license for their code. I'm confused as to what this means. Is the CHARMM-GUI MIT licensed or is the code openly available under MIT license? It is important to state clearly under which conditions the code is available because to my knowledge the actual code is not publicly available. If it is, I'd recommend mentioning it more clearly in the manuscript.

[Response 9] We are sorry that we referenced the MIT license in error. To summarize, though our server-side code is closed source, every molecular model a user builds on CHARMM-GUI includes all generated CHARMM scripts that were used in modeling. CHARMM-GUI's full licensing terms are shown here (https://charmm-gui.org/?doc=license_cgui):

“CHARMM-GUI is free for academic and governmental use, but not for commercial use or any collaboration with commercial entities.”

These terms apply to the usage of our web service CHARMM-GUI and to any output files we provide (e.g., CHARMM scripts). Anyone is free to modify and distribute copies of our provided CHARMM scripts for non-commercial purposes (simply because CHARMM, which we do not own, is not commercially free).

However, the new repository (https://github.com/charmm-gui/mca_scripts) for scripts used in preparation of this manuscript includes the MIT license.

Thank you again for your constructive suggestions!

Reviewer 3

We thank the Reviewer for the positive comments and suggestions. We revised the manuscript according to the Reviewer's comments as much as we could, and our specific responses are given below.

Comments

In this manuscript Kern *et al.* introduce a novel tool in CHARMM-GUI to enable the assembly of heterogeneous multicomponent systems and input preparation using PBC, which they validate using six model systems. I consider this a step forward in the applicability of the platform thereby enhancing the versatility of the CHARMM-GUI. However, there are some points that in my view require the attention of the authors.

[Comment 1] The nomenclature is not well defined, e.g. NP in the introduction, EO40EE37, NPAT, MEMB etc. The authors should either define all terms beforehand or introduce a table with the definitions in the manuscript.

[Response 1] We added the following clarifications based on the Reviewer's comment.

Introduction section (p1)

<Before>

Preparing MD simulation systems typically requires determining the model size and composition, solving an NP hard packing problem

<After>

Preparing MD simulation systems typically requires determining the model size and composition, solving a challenging packing problem

Results & Discussion section 1, STEP 3

<Added>

All membrane lipids built by *Membrane Builder* are given the segment identifier MEMB.

Results & Discussion section 1, STEP 5

<Before>

Finally, multiple MEMB segments (if any) are joined to a single segment, so that appropriate restraints can be generated in STEP 6.

<After>

Finally, any uploaded components with the segment ID MEMB and membrane generated by *Membrane Builder* (if any) are joined into a single segment, so that appropriate restraints can be generated in STEP 6.

Results & Discussion section 1, STEP 6

<Before>

For systems containing periodic nanomaterials or polymers, the generated NVT equilibration inputs use a multi-step scheme where protein restraints are progressively released. Inputs generated for NPT production use anisotropic pressure coupling, and NPAT production inputs use semi-isotropic pressure coupling with pressure applied only along the Z dimension.

<After>

For systems containing periodic nanomaterials or polymers, the generated NVT (constant number of particles, volume, and temperature) equilibration inputs use a multi-step scheme where protein restraints are progressively released. Inputs generated for NPT (constant number of particles, pressure, and temperature) production use anisotropic pressure coupling, and NPAT (constant number of particles, pressure, membrane area, and temperature) production inputs use semi-isotropic pressure coupling with pressure applied only along the Z dimension.

Results & Discussion section 2 (p2)

<Before>

At 5% v/v, proteins are very likely ($p > 0.5$) to contact HAP and EO₄₀EE₃₇

<After>

At 5% v/v, proteins are very likely ($p > 0.5$) to contact hydroxyapatite (HAP) and polyethylene oxide-poly(ethylene) (EO₄₀EE₃₇)

[Comment 2] The authors mention that “no free software package can handle all steps of model preparation without significant experimental knowledge, manual intervention, or use of ad hoc scripts” and underline at several locations that they aim to lower the entrance barrier for newbies in the field. I consider that the tone of these affirmations should be tuned down a bit. While I agree with the authors that at times a lower barrier might be useful, the user must have fundamental knowledge of their system, what they want to achieve and what the individual methods employed do. Hence, the user should not be encouraged to use the proposed package as a black box, which is what the strength of these sentences underline.

[Response 2] To avoid ambiguity and overstating our case, we have revised the following sentence from our Introduction.

<Before>

However, except for a few notable model types, no free software package can handle all steps of model preparation without significant experimental knowledge, manual intervention, or use of *ad hoc* scripts.

<After>

Although all software packages require familiarity with fundamentals of molecular modeling, there are still opportunities to lower the entry barrier to entry for modeling systems with multiple proteins and/or metabolites of interest at all-atom resolution.

[Comment 3] Please cite the appropriate papers for all the mentioned software, e.g. PACKMOL, cellPACK, LipidWrapper etc.

[Response 3] PACKMOL²²⁻²⁴, cellPACK^{10,11}, LipidWrapper²⁵, etc. are all cited at the beginning of paragraph 2 in the Introduction. If the citations shown below are inappropriate, please suggest others.

10. T. Johnson, G. *et al.* 3D molecular models of whole HIV-1 virions generated with cellPACK. *Faraday Discuss.* **169**, 23–44 (2014).
11. Johnson, G. T. *et al.* cellPACK: A Virtual Mesoscope to Model and Visualize Structural Systems Biology. *Nat. Methods* **12**, 85–91 (2015).
22. Martínez, L., Andrade, R., Birgin, E. G. & Martínez, J. M. PACKMOL: A package for building initial configurations for molecular dynamics simulations. *J. Comput. Chem.* **30**, 2157–2164 (2009).
23. Schott-Verdugo, S. & Gohlke, H. PACKMOL-Memgen: A Simple-To-Use, Generalized Workflow for Membrane-Protein–Lipid-Bilayer System Building. *J. Chem. Inf. Model.* **59**, 2522–2528 (2019).
24. Soñora, M., Martínez, L., Pantano, S. & Machado, M. R. Wrapping Up Viruses at Multiscale Resolution: Optimizing PACKMOL and SIRAH Execution for Simulating the Zika Virus. *J. Chem. Inf. Model.* **61**, 408–422 (2021).
25. Durrant, J. D. & Amaro, R. E. LipidWrapper: An Algorithm for Generating Large-Scale Membrane Models of Arbitrary Geometry. *PLOS Comput. Biol.* **10**, e1003720 (2014).

[Comment 4] Please annotate/label the steps to be followed also in Fig. 1.

[Response 4] There are many steps to follow to produce the systems shown in Figure 1. These steps are described in the Methods section and the video demos. We doubt that there is enough room in the figure for such annotation. Please note that miscellaneous system details can be found in Supplemental Information **Table S1**.

[Comment 5] Given that the input still requires quite some manipulation of the files, it would be useful to include a concrete example, including scripts, clear commands/snapshots in the SI.

[Response 5] Miscellaneous file manipulation scripts are documented at https://github.com/charmm-gui/mca_scripts, which we also mention in a new Code Availability section.

[Comment 6] Typo “polling”

[Response 6] This is not a typo.

[Comment 7] At times more explanation is needed, e.g. “first letter corresponds to the segment type (P, D, R, H, C, W)” – what do these letters concretely represent? It would be useful to double check and clarify the explanations of Step 1.

[Response 7] To clarify the abbreviations, we amended the sentence in Results & Discussion (*section 1, STEP 1 (p3)*).

<Before>

Each input segment is then re-written to a PSF such that the first letter corresponds to the segment type (P, D, R, H, C, W)

<After>

Each input segment is then re-written to a PSF such that the first letter corresponds to the segment type (protein: P, DNA: D, RNA: R, heterogen: H, carbohydrate: C, water: W)

[Comment 8] Step 2: It is unclear how the sasa is calculated. Is this intrinsically implemented or are there additional steps the user needs to assess? Also, why is achieving a volume fraction higher than 30% a trial-and-error process and how can this be avoided?

[Response 8] We are sorry that we referenced SASA in error. The intended term is “solvent-accessible volume”, whose calculation is described in Results & Discussion, section 1 STEP 1 (p2). In summary, a component’s solvent-accessible volume is calculated by calculating its volume with atomic radii increased by 1.4 Å and subtracting the molecular volume.

Since SASA was mentioned in the caption for Figure 6, it has been revised as follows.

<Before>

protein volume includes the solvent accessible surface area (SASA).

<After>

protein volume includes the solvent accessible volume, as described in STEP 1.

[Comment 9] Before figure 5 the authors mention that the CG representation is replaced with all-atom models. How is this concretely done? Additional details are required.

[Response 9] To clarify the process, we made the following changes to Results & Discussion section 1.

STEP 1

<Before>

To facilitate membrane building, the uploaded coordinates are used to determine the volume of the molecular regions that would be located within, above, and below a membrane centered at $Z=0$ with a hydrophobic thickness of 24 Å.

<After>

To facilitate membrane building, the uploaded coordinates are used to determine the volume and center of mass of the molecular regions that would be located within, above, and below a membrane centered at $Z=0$ with a hydrophobic thickness of 24 Å.

STEP 2

<Before>

After dynamics, the CG models are replaced with all-atom models and collisions are minimized with up to 7 iterations of the greedy conformation search (see Supporting Information **Algorithm 1**).

<After>

Solvated components are initially positioned by applying a random rotation and aligning the component’s center of mass with the center of its corresponding CG particle. For membrane components, a root-mean-squared best-fit alignment is performed between the

resulting CG coordinates and the reference coordinates calculated in STEP 1; the Z axis rotation and X/Y translation results of this best fit are applied to the all-atom model. Then, collisions are minimized with up to 7 iterations of the greedy conformation search (see Supporting Information **Algorithm 1**).

[Comment 10] Protein-protein and protein-membrane contacts

- a. As the authors choose to reproduce already published results as validation, they should compare their results to those in Ref. 59.
- b. I am curious why in Fig 6 for the second row the probability for Ubiquitin-membrane interaction decreases at 10% and then increases at 30% (comparable to 0%). Since the last paragraph is very descriptive, I would appreciate an interpretation of the results also in light (and added value) to Ref. 59.

[Response 10] We added the following paragraph at the end of the section to address the Reviewer's comment.

<Added>

Although we modeled the same CHL1/POPC/PSM membrane and proteins as Nawrocki *et al.*,⁶¹ we observed overall higher protein aggregation. For example, their contact fraction between villin and other villin molecules (villin-villin contacts) ranges between 25-30% whereas ours is between 27-73%. Although we both report overall higher protein contact probabilities with increasing protein concentrations, there are exceptions to this trend: Nawrocki *et al.* show a statistically insignificant decrease in villin-villin contact between 5% and 10% v/v. In contrast, only our protein G-protein G contact fraction decreases between 5% and 10%. Notably, they used the CHARMM36 FF with modified ion parameters, and they also decreased protein aggregation by multiplying protein-water Lennard-Jones (LJ) potentials by 1.09, whereas we used the CHARMM36m FF that uses an updated CMAP potential to reduce left-handed α -helix formation but does not directly address protein-water interaction strength by default.⁴⁸ Due to these methodological differences, a direct comparison of our results with those of Nawrocki *et al.* is challenging. Further study could see if reintroducing water interaction scaling reconciles this discrepancy.

We further compared our protein-membrane interaction results with Nawrocki *et al.* in the second paragraph of the same section.

<Added>

Indeed, all protein-CHL1/POPC/PSM contacts surprisingly decrease between 5–10% v/v and increase between 10–30% v/v, suggesting a strong preference for protein-protein contacts that is only overcome by increased protein crowding. The higher protein-membrane contacts at 5% v/v can be explained by the lack of opportunities for protein cluster formation⁶¹. It should be noted that this trend has a small magnitude: even the highest protein-CHL1/POPC/PSM contact fraction (7.6% for ubiquitin-membrane at 30% v/v) is lower than the lowest protein-membrane contact fraction with other membranes (14.5% for protein G-axolemma at 5% v/v). We believe more simulation replicas are required to establish the statistical significance of this trend reversal. The overall low protein-CHL1/POPC/PSM contacts and high contacts with the other membranes we simulated are consistent with Nawrocki *et al.*'s finding that proteins are thermodynamically excluded from membranes with no charged lipids⁶¹.

[Comment 11] All the simulation details for all the different systems the authors use as validation should be described, including force-fields, simulation length, water models, ions etc. This is very important for the reproducibility study as the authors directly aim to compare to something.

[Response 11] We amended **Table S1** to include the caption below. Other simulation details—number of each component type, initial box dimensions, and simulation runtime—are included in this table.

Table S1 caption

<Added>

The TIP3P water model and 0.15 M KCl are used in all aqueous systems. All proteins and lipids are modeled with the CHARMM36(m) FF; polymers and CO₂ are modeled with CGenFF; mica and hydroxyapatite are modeled with the INTERFACE FF.

[Comment 12] Figure 7: it is unclear why the displacement in the bottom leaflet (D1) is slower. The authors also mention that the results are comparable to experiments but there is no mention of what type of experiments these are.

[Response 12] In all simulations containing Mica, the bottom POPC leaflet is the one closest to Mica. The difference between the systems is the thickness of the water layer separating Mica and POPC's bottom leaflet ($D_1 = 1$ nm, $D_2 = 2$ nm, $D_3 = 3$ nm). Thus, D_1 is the system that provides the greatest interaction between Mica and POPC. To quantify this interaction, we used the CHARMM `INTER` function to calculate interaction energies via the formula $\Delta E_{\text{inter}} = E_{\text{both}} - (E_{\text{POPC}} + E_{\text{Mica}})$ for the final frame of each simulation. CHARMM energy calculations automatically separate different energy terms; the VDW (ΔE_{VDW}) and electrostatic (ΔE_{elec}) terms are reported in **Table R1** below.

Table R1. Interaction energies between POPC and Mica in kcal/mol.

	$D_{1\text{ nm}}$	$D_{2\text{ nm}}$	$D_{3\text{ nm}}$
ΔE_{total}	-88.36	-4.65×10^{-3}	0
ΔE_{VDW}	-29.34	-3.95×10^{-3}	0
ΔE_{elec}	-59.01	-7.08×10^{-4}	0

We believe this attraction hinders lateral diffusion of POPC by increasing friction between POPC and water.

To mention what type of experiments were used in our comparison, we amended Results section 3 as shown below:

<before>

Experimental values of the POPC diffusion coefficient in a bilayer are $\sim 7 \mu\text{m}^2/\text{s}^2$, implying that a bilayer on support behaves like free-standing lipids when the water thickness is more than 2 nm. In contrast, when the thickness of the water layer is less than 2 nm, there are strong interactions between the support and the lipids, which reduces the diffusion coefficient of the lipid in the lower leaflet of the SLB. This finding is consistent with

experimental measurements, which have shown that the thickness of the water layer is in the range of 1~2 nm^{62,63}.

<after>

Experimental values of the POPC diffusion coefficient measured by raster image correlation spectroscopy (RICS) of a lipid-like probe in a GUV bilayer are $\sim 7 \pm 3 \mu\text{m}^2/\text{s}$ ⁶⁴, implying that a bilayer on support behaves like free-standing lipids when the water thickness is more than 2 nm. In contrast, when the thickness of the water layer is less than 2 nm, there are strong interactions between the support and the lipids, which reduces the diffusion coefficient of the lipid in the lower leaflet of the SLB. This finding is consistent with water thickness of dimyristoylphosphatidylcholine (DMPC) SLBs measured with specular reflection of neutrons, which found that water thickness is in the range of 2~4 nm⁶⁵.

References

64. Gielen, E. *et al.* Measuring Diffusion of Lipid-like Probes in Artificial and Natural Membranes by Raster Image Correlation Spectroscopy (RICS): Use of a Commercial Laser-Scanning Microscope with Analog Detection. *Langmuir* **25**, 5209–5218 (2009).
65. Johnson, S. J. *et al.* Structure of an adsorbed dimyristoylphosphatidylcholine bilayer measured with specular reflection of neutrons. *Biophysical Journal* **59**, 289–294 (1991).

[Comment 13] Diffusion of CO₂: More details on the system and comparison to existing data (simulation, experiment?) is required. The results are descriptive and lacking interpretation; hence it is unclear why these results are relevant and/or reproduce correctly the relevant data.

[Response 13] To address this comment, we added the following section to the beginning of Supplemental Information.

<Added>

CO₂ Diffusion

We used the methods in Im and Roux's 2002 study¹ to calculate position dependent diffusion coefficients along the Z axis ($D(z)$) for embedded CO₂ with lag times (τ) from 1–10 ns (**Figure S7**). Mean and standard errors were calculated from the same bin positions across different simulation replicas, and diffusion in the polymer center (**Table S4**) was calculated by averaging all observations within bins located between $-15 \text{ \AA} < z < 15 \text{ \AA}$.

We found that $D(z)$ varies substantially by Z position, even within the polymer center, as CO₂ molecules find defects in the polymer structure where local diffusion is increased on short time scales. The mean of diffusion across bins in the polymer center was calculated to be almost 2x larger for PET₉₅ ($7.8 \pm 2.2 \text{ cm}^2/\text{s} \times 10^{-8}$) than PEF₉₅ ($4.0 \pm 0.3 \text{ cm}^2/\text{s} \times 10^{-8}$), possibly due to there being more defects within the PET₉₅ structure. Of the studies of CO₂ diffusion in PET we surveyed (**Table S4**), the most similar rate is in a simulation ($2.43 \text{ cm}^2/\text{s} \times 10^{-7}$ at 25 °C) which used lag times on the order of picoseconds. An experimental study² reported much slower CO₂ diffusion ($7.2 \pm 2 \text{ cm}^2/\text{s} \times 10^{-11}$ and $2.2 \pm 0.4 \text{ cm}^2/\text{s} \times 10^{-9}$ for PEF and PET, respectively), however their finding that CO₂ diffuses more slowly through PEF than PET is consistent with ours. That study measured permeation more directly via changes in atmospheric pressure across a macroscopic plastic barrier, whereas our study measures diffusion by tracking individual particles in a microscopic barrier.

To compare with other diffusion studies, it should be noted that the scale of $D(z)$ depends on τ ; the smaller timescales of simulations necessitate smaller τ values, which tends to overestimate long-term diffusion compared to experiments. The utility of simulation studies in this domain is thus by comparison of relative diffusion rates, rather than calculation of absolute diffusion rates.

A new reference to the SI was added to the section “Diffusion of CO₂ through polymer membranes” in the main text.

<Before>

Notably, between 666–1333 ns, the CO₂ density in near the polymer center (0–12 Å) was higher for PET₉₅ than for PEF₉₅, and the CO₂ density near the polymer periphery (15–37 Å) was lower for PET₉₅ than for PEF₉₅ in the same time range, indicating that PEF₉₅ is overall more resistant to CO₂ diffusion.

<After>

Notably, between 666–1333 ns, the CO₂ density in near the polymer center (0–12 Å) was higher for PET₉₅ than for PEF₉₅, and the CO₂ density near the polymer periphery (15–37 Å) was lower for PET₉₅ than for PEF₉₅ in the same time range, indicating that PEF₉₅ is overall more resistant to CO₂ diffusion. Indeed, CO₂ diffusion measured in the polymer center was nearly twice as high for PET₉₅ ($7.8 \pm 2.2 \text{ cm}^2/\text{s} \times 10^{-8}$) versus PEF₉₅ ($4.0 \pm 0.3 \text{ cm}^2/\text{s} \times 10^{-8}$), as described in **Supplemental Information: CO₂ Diffusion**.

We added the following figure, table, and references to the Supplemental Information.

<Added>

Figure S7. Position dependent diffusion profiles of CO₂ through PET₉₅ and PEF₉₅ in its approximate center. The purple and green lines were calculated from the equation $D = \text{MSD}(\tau) / (2\tau)$ with lag times of 2 and 5 ns, where D is diffusion, $\text{MSD}(\tau)$ is the measured mean square displacement of CO₂ when using a given lag time τ . The blue lines result from linear regression of the same equation with all lag times (at 0.1 ns intervals) from 1.0 to 10.0 ns. Error bars represent the standard error of the mean across all simulation replicas within each bin.

Table S4. Selected diffusion constants in this and other studies. Error/confidence intervals are shown where included in the original study.

Reference	Study Type	Polymer	D_{CO_2} (cm ² /s)	Temperature
2	Experiment	PEF	$(7.2 \pm 2) \times 10^{-11}$	35 °C
2	Experiment	PET	$(2.2 \pm 0.4) \times 10^{-9}$	35 °C
3	Experiment	PET	1×10^{-9}	STP
4	Simulation	PET	2.43×10^{-7}	25 °C
This study	Simulation	PEF	$(1.7 \pm 0.1) \times 10^{-7}$	25 °C
This study	Simulation	PET	$(1.5 \pm 0.1) \times 10^{-7}$	25 °C

References

1. Im, W. & Roux, B. Ions and Counterions in a Biological Channel: A Molecular Dynamics Simulation of OmpF Porin from Escherichia coli in an Explicit Membrane with 1M KCl Aqueous Salt Solution. *J. Mol. Biol.* 319, 1177–1197 (2002).
2. Burgess, S. K., Kriegel, R. M. & Koros, W. J. Carbon Dioxide Sorption and Transport in Amorphous Poly(ethylene furanoate). *Macromolecules* 48, 2184–2193 (2015).
3. Okuji, S. et al. Surface modification of polymeric substrates by plasma-based ion implantation. *Nucl. Instrum. Methods Phys. Res. Sect. B Beam Interact. Mater. At.* 242, 353–356 (2006).
4. Liao, L.-Q., Fu, Y.-Z., Liang, X.-Y., Mei, L.-Y. & Liu, Y.-Q. Diffusion of CO₂ Molecules in Polyethylene Terephthalate/Poly lactide Blends Estimated by Molecular Dynamics Simulations. *Bull. Korean Chem. Soc.* 34, 753–758 (2013).

[Comment 14] The authors stress that the study presents an “automated and versatile procedure for building”. I recommend softening these types of statements as quite some manipulation is required, a fact also acknowledged by the authors in the next paragraph.

[Response 14] We reworded the conclusion to reflect the Reviewer’s comment.

<Before>

This work presents an automated and versatile procedure for building simulation-ready, atomistic models containing heterogeneous molecular components via *Multicomponent Assembler* (MCA) in CHARMM-GUI. Automation is achieved by assigning component types that determine the packing strategy and using a greedy packing algorithm whose initial configuration is generated from coarse-grained simulation.

<After>

This work presents a guided procedure for building simulation-ready, atomistic models containing heterogeneous molecular components via *Multicomponent Assembler* (MCA) in CHARMM-GUI. Initial positioning is facilitated by assigning component types that determine the packing strategy and using a greedy packing algorithm whose initial configuration is generated from CG simulation.

[Comment 15] General comments

- a. The figures would benefit from larger labels, thicker lines, and a better choice of colors. Generally, on a printed version of the manuscript most lines are not visible or indistinguishable, the labels and legends are too small.

- b. In Fig. S4 It would be useful to mention if these are representative structures and if so, how they were chosen. If clustering was performed, then how?

[Response 15] The structures in **S4** were identified by viewing the simulation trajectories in VMD. No clustering was performed. The structures were chosen only to provide an example of a long-lived contact event. The following statement was added to the **Figure S4** caption.

<Added>

These representative structures were identified visually.

We furthermore increased line thickness and font size in most figures. Thank you!

REVIEWER COMMENTS

Reviewer #1 (Remarks to the Author):

I think the authors did a thorough job responding to the questions from the previous round of review, including questions from the other reviewers.

Reviewer #2 (Remarks to the Author):

see comments attached

Reviewer 1

We thank the Reviewer for the positive comments. Since there are no suggestions to revise the manuscript, we have no specific responses.

Reviewer 2

We thank the Reviewer for the positive comments and suggestions. We revised the manuscript according to the Reviewer's comments as much as we could, and our specific responses are given below.

Comments

In this paper, Kern et al. present and demonstrate the algorithm, which underlies the CHARMM-GUI Multicomponent Assembler (MCA). The MCA is a webserver-based workflow for assembling coordinates and generating simulation input parameters necessary to start molecular dynamics simulations of complex systems. The MCA workflow is the latest among a long list of CHARMM-GUI tools and specifically implements the following new features: (1) a PBC-aware packing algorithm; (2) a generalized template-based solvent approach for arbitrary mixtures of solvent molecules; (3) a PBC-aware utilization of a periodic membrane.

[Comment 1] A program or workflow for packing multicomponent systems is in itself not novel and many tools have been presented in the literature. The introduction gives an adequate overview of the most popular tools in the field of biomolecular simulations. A minor point of critique is the lack of tools from the material science side – especially for generating systems containing polymers, which is one of the test cases presented later. For example, moltemplate, polyply, pysimm, are all programs that could be used to create example case 4.

[Response 1] While it may indeed be the case that our example case 4 (polymer slab + proteins in separate solvated region) can be created by moltemplate, polyply, and pysimm, these programs and their documentation or example codes do not appear to contain a workflow designed to facilitate such a system. Rather, they use a more general *scripting* approach that relies on the user to design the workflow with the functions provided by each program.

In a similar sense, CHARMM—which implements a general modeling language—can be used to build the example case 4, but it required years of development for us to design the MCA workflow using CHARMM that can handle all of the example cases we presented. Note that the result of this workflow is the main content of our manuscript. If we were to include moltemplate, polyply, or pysimm in a benchmark, we would first need to design a workflow for each program. Besides the considerable efforts this would require, it is unlikely we would be able to present a fair comparison with MCA, since we are not experts in using those programs for workflow design.

For example, the work of J. A. Stevens, *et al.* (2023), referenced by the reviewer, uses experimentally obtained component positions to construct a whole cell model that is assembled with Polyply, Martinize2, TS2CG, and Bentopy (an unpublished package). While this construction is impressive and we await Bentopy's publication with anticipation, a direct comparison with MCA appears to be challenging.

As the Reviewer suggested, however, we performed the benchmark study shown below for MCA and PACKMOL using biomolecules with varying asphericity. The Syn3A model includes long chromosomes that cannot be modeled well by either MCA or PACKMOL because both programs treat molecules as rigid bodies, as explained in **Response 7**. Therefore, we chose to instead model a smaller set of representative molecules and added the following new Section 5 in Results & Discussion.

[Response 1R]

The authors' argument is valid, which is why I considered this problem to be a minor point. I was not suggesting any comparison to Bentopy (indeed an unpublished package) or building the cell as a form of benchmark. I'm happy that the authors have included a more detailed benchmark comparison with PACKMOL, which is the most similar program capability-wise. Comparing the MCA capabilities to other programs is also a valuable addition. However, I've some comments on this addition (see below).

<Added>

5. Comparison with other programs

Several existing programs facilitate densely packed macromolecular system modeling. For example, PACKMOL and cellPACK can pack molecules into complex shapes such as budding vesicles and viral capsids^{10,11,24}; polyply and pysimm can model long polymer chains (e.g., DNA/RNA) densely packed with proteins in a cytoplasm^{26,29}. Moltemplate can also model long polymers, but collision detection must be handled externally⁷⁰. **Table S2** summarizes our own analysis on the capabilities of each program in multicomponent assembly, which is elaborated below.

Pysimm is a Python API that facilitates modeling and simulation with LAMMPS. It can read molecular structures in several formats including PDB, XYZ, and MOL/MOL2. It can construct LAMMPS-compatible topologies and supports creating and positioning long polymers. Pysimm includes many functions that delegate common simulation tasks to LAMMPS, such as molecular dynamics and minimization. Pysimm is similar to pyCHARMM⁷¹ embedding CHARMM functionality in a python framework.

Moltemplate is designed for CG simulations, so that third-party molecular builder tools are required to construct all-atom models of biomolecules such as proteins, RNA, and DNA.

I'm a little confused by this sentence, since Moltemplate states on their website: "Moltemplate was intended for building custom coarse-grained molecular models, but it can be used to prepare realistic all-atom simulations as well." As far as material science systems such as polymer melts, surfaces, and solutions, are concerned, the moltemplate website lists quite some examples for generating these.

As described in our recent publication³⁰, the initial polymer configurations generated by CHARMM-GUI *Polymer Builder* exhibit structures similar to fully relaxed configurations, allowing for direct production runs without additional equilibration simulations. However, when utilizing moltemplate's linear stacking method, additional equilibration simulation of tens to hundreds of nanoseconds are required even after the initial structure formation.

I find the above paragraph a little confusing, though I agree with the content. My suggestion would be to flip the two sentences around. Something along the lines of

"When utilizing moltemplate's linear stacking method to generate initial polymer conformations, long equilibration simulation of tens to hundreds of nanoseconds are required before the melt is well relaxed. In contrast, the CHARMM-GUI Polymer Builder creates initial structures similar to fully relaxed configurations, allowing for direct production runs without long equilibration simulations."

Polyply has been developed for CG simulations using the Martini force field and recently extended to support all-atom models for certain long polymers and carbohydrates. However, similar to Moltemplate, third-party software is required for biomolecules.

This statement is inaccurate. Polyply has been developed to generate topologies and coordinates irrespective of the force field or resolution. Already the first published version supports all-atom and CG polymers. For example, the publication (<https://www.nature.com/articles/s41467-021-27627-4#Sec2>) contains benchmarks for polymer melts at the all-atom level as well as polymers at Martini resolution (see Figure 3b). Note that Table S2 should also be updated accordingly. According to the polyply GitHub page

(see: https://github.com/marrink-lab/polyply_1.0/blob/master/LIBRARY.md),

it currently supports the following forcefields: Martini 2/3, Amber variant ParmBsc1 for DNA, GROMOS (2016H66, and 53A6), OPLSAA. For the sake of not having unnecessarily long discussions of programs, I would propose to rephrase this paragraph along these lines of:

“Polyply has been developed to generate input files and starting coordinates for polymeric molecules independent of the level of resolution (e.g. Martini but also all-atom). However, it does not perform rigid body packing of large molecules (e.g. proteins) and requires third-party software for certain biomolecules, similar to Moltemplate.”

PACKMOL has been developed for finding collision-free rigid body packing solutions of arbitrary molecules in many non-periodic geometries. It implements an intuitive scripting format for describing the geometrical requirements and can read input coordinates from PDB, tinker, xyz, and moldy formats. PACKMOL has quick runtimes with stable performance when using default settings. This makes it an ideal candidate for benchmark comparison with MCA. As both MCA and PACKMOL treat molecules as rigid bodies, neither software package is well-suited to mixing long polymers with proteins.

See language edits in the part below.

To test whether our packing approach can lead to more dense packing and lower runtimes **even though** the CHARMM executable is not optimized for packing, we selected two sets of macromolecules shown in **Table S3**: one where all molecules have asphericity between 0.07 – 0.15 (easy) and the other where asphericity varies between 0.02 – 0.48 (hard). We then ran 12 replicas of both sets through PACKMOL and MCA with volume fraction (v/v) starting from 10% and increasing by 1% until all 12 replicas failed at the same v/v. As shown in **Figure S6**, MCA's runtimes were longer for **low-density** systems, and shorter for **high-density** systems. The maximum v/v achieved by PACKMOL was 30% (easy) and 18% (hard), and it was 41% (easy) and 23% (hard) for MCA. These results show that our approach can improve packing performance for periodic geometries when the system density is high.

The following elaboration of our benchmark environment was added to Methods.

<Added>

MCA and PACKMOL Packing Benchmark Environment

Computing environment

We used one Dell XPS 8900 workstation with an Intel Core i7-6700 CPU (4 cores) and 8 GB of installed RAM. On the Ubuntu 22.04.2 LTS operating system, we installed PACKMOL 20.14.2 and CHARMM 48a1 (commit ID cb36cf5c6) from CHARMM's development branch.

To set an appropriate number of simultaneous tests to run on the workstation, we observed CPU and memory usage of solo tests and determined that while PACKMOL and CHARMM

consistently consumed 1 CPU at a rate near 100%, PACKMOL's memory usage was uniformly below 9 MB, whereas CHARMM tests used 3.5 GB. To make the most of the workstation's resources, we thus ran 4 simultaneous jobs when testing PACKMOL and 2 when testing CHARMM.

In all tests, target system densities were achieved by varying total system volume while keeping the number of molecules constant. All code used to run our benchmark can be found at https://github.com/charmm_gui/mca_scripts.

MCA

One MCA job was created on CHARMM-GUI for each set of macromolecules shown in **Table S3**. After selecting the number of molecules and a 10% volume fraction, we performed packing in the web GUI and downloaded the projects from the solvent options page without generating a solvent. Each macromolecule set's project directory was copied to create 12 replicas (24 total). The directories were copied again to create one directory for each tested volume fraction at 1% intervals starting from 10% and increasing until all replicas failed at the same v/v. Random number generator (RNG) seeds were set automatically from CPU time.

PACKMOL

To avoid giving MCA an undue advantage, hydrogen atoms were stripped from PACKMOL's input PDB files, as MCA ignores hydrogen atoms when checking for collisions. Similarly, the collision tolerance was set to 2.5 Å to match MCA's tolerance. RNG seeds were set automatically from CPU time. Default values were used for all other settings. As with MCA, 12 replicas of each macromolecule set were tested at 1% v/v intervals starting from 10% and increasing until all replicas failed at the same v/v.

The following figure and tables were added to Supplemental Information.

<Added>

Table S2. Analysis of various programs for multicomponent molecular assemblies. "Special script" refers to a scripting language created specifically for a given modeling program. Topology preparation is "automatic" if individual molecule topologies can be inferred or read from a database and "manual" if they must be provided separately by the user.

	MCA	Pysimm	Moltemplate	PACKMOL	Polyply
User Interface	GUI or CHARMM script	Python API	Special script	Special script	Special script
Supported Simulation Programs	GROMACS CHARMM OpenMM Amber GENESIS Desmond	LAMMPS CASSANDRA	LAMMPS	N/A	GROMACS
Supported Force Fields	CHARMM AMBER	CHARMM AMBER GAFF DREIDING PCFF	OPLS COMPASS MARTINI GAFF DREIDING	N/A	MARTINI GROMOS AMBER OPLS
Topology Preparation	Automatic	Automatic	Manual	N/A	Manual

Table S3. Molecules used in benchmark. 10 copies (easy) and 8 copies (hard) of each molecule were packed into cubic geometries.

Test Name	PDB ID	Asphericity	# Residues	Volume (\AA^3)
Easy	1ubq	0.07	76	9.37×10^3
	1vii	0.14	36	4.45×10^3
	3gb1	0.15	56	6.62×10^3
Hard	1mjc	0.02	69	7.84×10^3
	3gb1	0.14	56	6.62×10^3
	1vii	0.15	36	4.45×10^3
	6y3g	0.41	87	2.15×10^4
	2hac	0.48	60	7.78×10^3

Figure S6. Comparison of performance between PACKMOL and MCA. (A) Mean \pm standard error runtime of packing tasks at a given volume fraction (% v/v). The last point in each line is the v/v at which all packing attempts failed. (B) The fraction of packing attempts that fail. Only the lowest v/v resulting in 100% failure is shown for each combination of program and molecule set.

[Comment 2] The authors identify several shortcomings of the existing tools that they aim to resolve with the MCA. Those shortcomings are the lack of PBC-aware packing, “significant experimental knowledge, manual intervention, or use of ad hoc scripts” to be able to generate starting coordinates and simulation parameters, as well as the need for “significant post-processing”. The lack of PBC-aware packing and the fact that potentially more than one program/script has to be used to generate both simulation input parameters and coordinates, I consider valid shortcomings of the existing tools mentioned. However, this paragraph contains some inappropriate generalizations. For instance, what constitutes “significant experimental knowledge” that the MCA does not require? What is “manual intervention” in the context of generating coordinates and topologies? Doesn’t the MCA also require the user to generate the

appropriate CHARMM input in multiple manual steps? To consider the paper for publication I strongly recommend working out more clearly what MCA brings to the table and rephrasing the previously mentioned generalizations or supporting them by references to literature.

[Response 2] To avoid ambiguity and overstating our case, we have revised the following sentence from Introduction.

<Before>

However, except for a few notable model types, no free software package can handle all steps of model preparation without significant experimental knowledge, manual intervention, or use of *ad hoc* scripts.

<After>

Although all software packages require familiarity with fundamentals of molecular modeling, there are still opportunities to lower the entry barrier for modeling systems with multiple proteins and/or metabolites of interest at the all-atom resolution.

[Response 2R]

The rephrased sentence is indeed more appropriate.

[Comment 3] Considering the widespread use of CHARMM and the CHARMM-GUI, it is of significant interest to the scientific community to have a tool such as MCA, which allows the combination of the different CHARMM-GUI modules in a consistent manner. Similar papers describing tools to set up MD simulations have already been published in the journal at hand.[2,4] However, it needs to be kept in mind that the new features the MCA offers are the assembly and packing of the different components. Protocols for generating proteins, membranes, or polymers have already been published previously.[5,6]

[Response 3] MCA indeed does not re-implement the procedures in *Polymer Builder* or *Input Generator* (now *FF-Converter*), but instead builds on and integrates with them.

[Response 3R]

It doesn't practically integrate them, does it? The user still has to go to a separate builder module and work through the respective workflow (that's already published). Only then can the system be assembled. However, this statement was intended as an observation rather than a point of critique.

We have added citations for TS2CG, polyply, pysimm, and Polymer Builder to Introduction.

<Before>

Many software applications have been developed to facilitate various steps of atomistic model preparation, including FFParm¹², FFTK¹³, SwissParam¹⁴, Antechamber¹⁵, CGenFF¹⁶⁻¹⁸, MATCH¹⁹, OpenFF²⁰, and CHARMM-GUI *Ligand Reader & Modeler*²¹ for FF preparation; PACKMOL²²⁻²⁴, cellPACK^{10,11}, LipidWrapper²⁵, and Soup²⁶ for molecular packing; and *FF-Converter*^{27,28} and ParmEd²⁹ for preparing and converting inputs for several simulation programs.

<After>

Many software applications have been developed to facilitate various steps of atomistic model preparation, including FFParm¹², FFTK¹³, SwissParam¹⁴, Antechamber¹⁵, CGenFF¹⁶⁻¹⁸, MATCH¹⁹, OpenFF²⁰, and CHARMM-GUI *Ligand Reader & Modeler*²¹ for FF preparation; PACKMOL²²⁻²⁴, cellPACK^{10,11}, TS2CG²⁵, LipidWrapper²⁶, and Soup²⁷ for molecular packing; polyply²⁸, pysimm²⁹, and *Polymer Builder*³⁰ for building and assembling

long polymer chains; and *FF-Converter*^{31,32} and ParmEd³³ for preparing and converting inputs for several simulation programs.

[Comment 4] At the same time there are several shortcomings of the current manuscript, which in my opinion need to be improved before the paper can be published:

A core improvement of MCA over existing tools is the PBC-aware packing algorithm. While the six benchmark cases are complex enough to support the claim of wide-spread applicability, the manuscript lacks proper benchmarks for the packing algorithm itself. For the highly concentrated protein systems, the authors choose three rather small and spherical proteins to pack. I'd argue that these proteins are therefore quite easy to pack. A proper benchmark should consider a wider set of diversly shaped proteins and assess the speed comparison to existing tools (take Packmol[7] for example), and the resilience (i.e., how often does it crash). A diverse set of proteins that is biologically relevant as well are the cellular proteins of the Syn3A minimal cell. Interestingly, the protein content of this cell is also about 30% of the volume [8], which is the upper limit for efficient packing using MCA. I also suggest reporting the sphericity or asphericity of the proteins.

[Response 4] As the Reviewer suggested, we performed the benchmark study shown in **Response 1** using MCA and PACKMOL using biomolecules with varying asphericity. The Syn3A model includes long chromosomes that cannot be modeled well by either MCA or PACKMOL. Therefore, we chose to instead model two smaller sets of representative molecules, as explained in **Response 1**.

[Response R4]

This new benchmark nicely demonstrates the power of MCA. Consistently achieving nearly 30% packing density is a quite good result. I would recommend moving this benchmark to the main paper. However, in light of the already rather long main paper, this suggestion is optional.

[Comment 5] Along the lines of the previous point, the authors mention in their demo video that the packing algorithm can fail if solution components are too close to the membrane or "bad" lipids are selected. A proper benchmark of how robust the packing algorithm is considering more difficult lipid membranes (e.g., containing largish glycolipids and larger transmembrane proteins) seems appropriate to substantiate their claim that MCA does not require significant manual intervention. Repeated trial and error for generating coordinates to me is not so different from having to use multiple scripts and can be rather daunting for non-expert users. This benchmark appears especially important as non-expert users appear to be one of the target groups for MCA.

[Response 5] We agree with the Reviewer that the level of knowledge about one's modeling goals and possible trial and error necessary to reconcile collisions between long glycolipids and solvated proteins can count as "manual intervention". Therefore, we rephrased our claims more modestly, as explained in **Response 2**.

[Response R5]

ok

[Comment 6] While it seems evident that the MCA can generate combinations of pre-build polymer systems and proteins, I wonder if it is possible to generate a protein solution in the presence of polymers. For example, a protein solution with crowders such as PEG is a frequently used cell mimetic. Can the MCA build such a system? Other tools such as moltemplate or polyipy could for example be used for such systems.

[Response 6] MCA itself is ill-suited to create this type of system, or even the polymer slab we used in the example case 4. Our polymer was created separately by *Polymer Builder*, then

uploaded to MCA. To discuss this limitation, we included the following paragraph in the new Section 6 in Results & Discussion, as explained more in **Response 7** (reprinted below).

<Added>

The rigid body packing used by MCA or PACKMOL tends to work well for packing problems involving macromolecules that are already fairly rigid, such as proteins, crystals, and short nucleic acid sequences. Packing long flexible molecules such as synthetic and nucleic acid polymers is unlikely to succeed because the molecule's ability to pack tightly with other components comes from its flexibility. In such cases, the polymers should be assembled by another method, such as the random walk algorithms used by pysimm[reference missing] and polyply[reference missing] or the CG Kuhn fragment equilibration method used by *Polymer Builder*³⁰.

[Response R6]

The limitation section is a good addition to the paper. However, when reading the paragraph, I understood that I could create a polymer solution with polymer builder and then add the proteins using MCA. But from the general response, I gather that would likely not work, or would it? I'm insisting on this class of systems because they are quite common in bio-physics / bio-materials (e.g. hydrogels, coacervates, antibody formulations, joint fluids, and many more).

[Comment 7] The complete lack of a discussion section, which also provides some limitations of the MCA, surprises me. I strongly recommend adding a discussion that sets into perspective the advantages of MCA in contrast to its limitations. In relation to existing tools, there are several limitations of the MCA in my opinion: Currently, it is not possible to easily implement molecules unknown to the CHARMM-GUI such as new lipids or polymers. While other tools may require some more knowledge, they are generally extendable, and implementing new molecules is well-documented. Furthermore, the support of an API makes it impossible to use MCA for automated pipelines or high-throughput workflows, which are more easily accessible in the other programs. For membranes and solutions, there exists a high throughput simulator. Can the MCA be connected to it in the future?

[Response 7] We eventually plan to create a web API to facilitate automated access to CHARMM-GUI. However, it will be a big project because all CHARMM-GUI modules would need to be modified to support this type of access. We do not currently have any plans to connect MCA to *High Throughput Simulator*, but we will consider the idea if we get the necessary funding and talent.

[Response R7]

Ok! I'm looking forward to when this is going to happen. See some language edits below on the added text section.

We agree that the current CHARMM-GUI workflow does not easily handle new lipid and polymer types, although CHARMM-GUI supports a diverse set of lipids and polymer monomers. MCA facilitates combining results of other CHARMM-GUI modules by accepting CHARMM-formatted topology, parameter, structure, and coordinate files produced by those modules. However, MCA does not require the files to be produced by CHARMM-GUI, only that they need to be in CHARMM format. Many other programs (e.g., psfgen for PSF; FFPParam, CGenFF, etc. for topologies and parameters; MDAnalysis etc. for CRD) are capable of producing these files. Regardless, users typically must be familiar enough with CHARMM file formats to resolve file reading errors.

In particular, to address this comment, we added the following new Section 6 in Results & Discussion.

<Added>

6. Limitations

Although there is a large library of molecules that can be handled by MCA via other CHARMM-GUI modules, incorporating molecules that are not available in CHARMM-GUI—such as new membrane lipids, synthetic polymer building blocks, or ligands unsupported by CGenFF—could be **extremely [too informal]** challenging for users.

Although users can download the CHARMM scripts used to generate a system, the scripts are often hundreds to thousands of lines long. While MCA allows users to upload their own CHARMM topologies (RTF) and parameters (PRM) for molecules not contained within the CHARMM36(m) or INTERFACE FFs, parameterizing molecules for the CHARMM FF is challenging. **However, many** other programs (e.g., psfgen for PSF⁷²; FFParm¹², CGenFF¹⁶⁻¹⁸, ~~etc. for topologies and parameters; MDAnalysis⁷³ etc. for CRD~~) are capable of producing **these topology and parameter files**.

[MDAnalysis is not really a library for generating coordinates but rather converting or reading them; I would just mention the programs for topology and parameter files.]

The rigid body packing used by MCA or PACKMOL tends to work well for packing problems involving macromolecules that are already fairly rigid, such as proteins, crystals, and short nucleic acid sequences. Packing long flexible molecules such as synthetic and nucleic acid polymers is unlikely to succeed because the molecule's ability to pack tightly with other components comes from its flexibility. In such cases, the polymers should be assembled by another method, such as the random walk algorithms used by pysimm and polyly or the CG Kuhn fragment equilibration method used by *Polymer Builder*³⁰.

[Comment 8] MCA when used together with other tools such as membrane builder can take several hours up to days in order to generate a morphology. To my knowledge, this performance is slower than that of comparable tools such as polyly for polymers or cellPack (especially with GPU acceleration) for molecular solutions. Of course, the advantages of MCA might well outweigh the disadvantages, but this bit deserves a discussion together with the limitations in point 4.

[Response 8] We agree that MCA and other CHARMM-GUI tools often have long runtimes for complex systems. However, we hope that our **Response 1** is sufficient to address this comment, in particular our inclusion of runtime results and discussion. We believe that CHARMM-GUI users are using CHARMM-GUI with some long runtimes as it is not easy to use other programs to accomplish a similar level of complexity without customized complex scripting. However, users can submit multiple jobs and return to them at their leisure through the use of Job Retriever—if they know their job's ID—or with the recently added “Job IDs” page of their user profile.

[Response R8]

The addition of run-time results nicely shows that MCA is faster than Packmol for most tasks. The overall run-time even for complex systems, is also more than acceptable. I'm satisfied with the discussion in the benchmark section.

[Comment 9] Finally, in their reporting summary, the authors give an MIT license for their code. I'm confused as to what this means. Is the CHARMM-GUI MIT licensed or is the code openly available under MIT license? It is important to state clearly under which conditions the code is available because to my knowledge the actual code is not publicly available. If it is, I'd recommend mentioning it more clearly in the manuscript.

[Response 9] We are sorry that we referenced the MIT license in error. To summarize, though our server-side code is closed source, every molecular model a user builds on CHARMM-GUI

includes all generated CHARMM scripts that were used in modeling. CHARMM-GUI's full licensing terms are shown here (https://charmm-gui.org/?doc=license_cgui):

“CHARMM-GUI is free for academic and governmental use, but not for commercial use or any collaboration with commercial entities.”

These terms apply to the usage of our web service CHARMM-GUI and to any output files we provide (e.g., CHARMM scripts). Anyone is free to modify and distribute copies of our provided CHARMM scripts for non-commercial purposes (simply because CHARMM, which we do not own, is not commercially free).

However, the new repository (https://github.com/charmm-gui/mca_scripts) for scripts used in preparation of this manuscript includes the MIT license.

Thank you again for your constructive suggestions!

[Response R9]

Thank you for clarifying this point and editing the checklist!

However, after inspecting the mentioned repository more closely, I realized that simulation parameters and starting structures of the different test cases are not provided. To improve reproducibility and adhere to the FAIR principles of science, I need to insist that these be made available as well. If the authors are worried about file sizes, they can upload the data to for example Zenodo instead of GitHub.

Reviewer 3

We thank the Reviewer for the positive comments and suggestions. We revised the manuscript according to the Reviewer's comments as much as we could, and our specific responses are given below.

Comments

In this manuscript Kern *et al.* introduce a novel tool in CHARMM-GUI to enable the assembly of heterogeneous multicomponent systems and input preparation using PBC, which they validate using six model systems. I consider this a step forward in the applicability of the platform thereby enhancing the versatility of the CHARMM-GUI. However, there are some points that in my view require the attention of the authors.

[Comment 1] The nomenclature is not well defined, e.g. NP in the introduction, EO40EE37, NPAT, MEMB etc. The authors should either define all terms beforehand or introduce a table with the definitions in the manuscript.

[Response 1] We added the following clarifications based on the Reviewer's comment.

Introduction section (p1)

<Before>

Preparing MD simulation systems typically requires determining the model size and composition, solving an NP hard packing problem

<After>

Preparing MD simulation systems typically requires determining the model size and composition, solving a challenging packing problem

Results & Discussion section 1, STEP 3

<Added>

All membrane lipids built by *Membrane Builder* are given the segment identifier MEMB.

Results & Discussion section 1, STEP 5

<Before>

Finally, multiple MEMB segments (if any) are joined to a single segment, so that appropriate restraints can be generated in STEP 6.

<After>

Finally, any uploaded components with the segment ID MEMB and membrane generated by *Membrane Builder* (if any) are joined into a single segment, so that appropriate restraints can be generated in STEP 6.

Results & Discussion section 1, STEP 6

<Before>

For systems containing periodic nanomaterials or polymers, the generated NVT equilibration inputs use a multi-step scheme where protein restraints are progressively released. Inputs generated for NPT production use anisotropic pressure coupling, and NPAT production inputs use semi-isotropic pressure coupling with pressure applied only along the Z dimension.

<After>

For systems containing periodic nanomaterials or polymers, the generated NVT (constant number of particles, volume, and temperature) equilibration inputs use a multi-step scheme where protein restraints are progressively released. Inputs generated for NPT (constant number of particles, pressure, and temperature) production use anisotropic pressure coupling, and NPAT (constant number of particles, pressure, membrane area, and temperature) production inputs use semi-isotropic pressure coupling with pressure applied only along the Z dimension.

Results & Discussion section 2 (p2)

<Before>

At 5% v/v, proteins are very likely ($p > 0.5$) to contact HAP and EO₄₀EE₃₇

<After>

At 5% v/v, proteins are very likely ($p > 0.5$) to contact hydroxyapatite (HAP) and polyethylene oxide-poly(ethylene) (EO₄₀EE₃₇)

[Comment 2] The authors mention that “no free software package can handle all steps of model preparation without significant experimental knowledge, manual intervention, or use of ad hoc scripts” and underline at several locations that they aim to lower the entrance barrier for newbies in the field. I consider that the tone of these affirmations should be tuned down a bit. While I agree with the authors that at times a lower barrier might be useful, the user must have fundamental knowledge of their system, what they want to achieve and what the individual methods employed do. Hence, the user should not be encouraged to use the proposed package as a black box, which is what the strength of these sentences underline.

[Response 2] To avoid ambiguity and overstating our case, we have revised the following sentence from our Introduction.

<Before>

However, except for a few notable model types, no free software package can handle all steps of model preparation without significant experimental knowledge, manual intervention, or use of *ad hoc* scripts.

<After>

Although all software packages require familiarity with fundamentals of molecular modeling, there are still opportunities to lower the entry barrier to entry for modeling systems with multiple proteins and/or metabolites of interest at all-atom resolution.

[Comment 3] Please cite the appropriate papers for all the mentioned software, e.g. PACKMOL, cellPACK, LipidWrapper etc.

[Response 3] PACKMOL²²⁻²⁴, cellPACK^{10,11}, LipidWrapper²⁵, etc. are all cited at the beginning of paragraph 2 in the Introduction. If the citations shown below are inappropriate, please suggest others.

10. T. Johnson, G. *et al.* 3D molecular models of whole HIV-1 virions generated with cellPACK. *Faraday Discuss.* **169**, 23–44 (2014).
11. Johnson, G. T. *et al.* cellPACK: A Virtual Mesoscope to Model and Visualize Structural Systems Biology. *Nat. Methods* **12**, 85–91 (2015).
22. Martínez, L., Andrade, R., Birgin, E. G. & Martínez, J. M. PACKMOL: A package for building initial configurations for molecular dynamics simulations. *J. Comput. Chem.* **30**, 2157–2164 (2009).
23. Schott-Verdugo, S. & Gohlke, H. PACKMOL-Memgen: A Simple-To-Use, Generalized Workflow for Membrane-Protein–Lipid-Bilayer System Building. *J. Chem. Inf. Model.* **59**, 2522–2528 (2019).
24. Soñora, M., Martínez, L., Pantano, S. & Machado, M. R. Wrapping Up Viruses at Multiscale Resolution: Optimizing PACKMOL and SIRAH Execution for Simulating the Zika Virus. *J. Chem. Inf. Model.* **61**, 408–422 (2021).
25. Durrant, J. D. & Amaro, R. E. LipidWrapper: An Algorithm for Generating Large-Scale Membrane Models of Arbitrary Geometry. *PLOS Comput. Biol.* **10**, e1003720 (2014).

[Comment 4] Please annotate/label the steps to be followed also in Fig. 1.

[Response 4] There are many steps to follow to produce the systems shown in Figure 1. These steps are described in the Methods section and the video demos. We doubt that there is enough room in the figure for such annotation. Please note that miscellaneous system details can be found in Supplemental Information **Table S1**.

[Comment 5] Given that the input still requires quite some manipulation of the files, it would be useful to include a concrete example, including scripts, clear commands/snapshots in the SI.

[Response 5] Miscellaneous file manipulation scripts are documented at https://github.com/charmm-gui/mca_scripts, which we also mention in a new Code Availability section.

[Comment 6] Typo “polling”

[Response 6] This is not a typo.

[Comment 7] At times more explanation is needed, e.g. “first letter corresponds to the segment type (P, D, R, H, C, W)” – what do these letters concretely represent? It would be useful to double check and clarify the explanations of Step 1.

[Response 7] To clarify the abbreviations, we amended the sentence in Results & Discussion (section 1, STEP 1 (p3)).

<Before>

Each input segment is then re-written to a PSF such that the first letter corresponds to the segment type (P, D, R, H, C, W)

<After>

Each input segment is then re-written to a PSF such that the first letter corresponds to the segment type (protein: P, DNA: D, RNA: R, heterogen: H, carbohydrate: C, water: W)

[Comment 8] Step 2: It is unclear how the sasa is calculated. Is this intrinsically implemented or are there additional steps the user needs to assess? Also, why is achieving a volume fraction higher than 30% a trial-and-error process and how can this be avoided?

[Response 8] We are sorry that we referenced SASA in error. The intended term is “solvent-accessible volume”, whose calculation is described in Results & Discussion, section 1 STEP 1 (p2). In summary, a component’s solvent-accessible volume is calculated by calculating its volume with atomic radii increased by 1.4 Å and subtracting the molecular volume.

Since SASA was mentioned in the caption for Figure 6, it has been revised as follows.

<Before>

protein volume includes the solvent accessible surface area (SASA).

<After>

protein volume includes the solvent accessible volume, as described in STEP 1.

[Comment 9] Before figure 5 the authors mention that the CG representation is replaced with all-atom models. How is this concretely done? Additional details are required.

[Response 9] To clarify the process, we made the following changes to Results & Discussion section 1.

STEP 1

<Before>

To facilitate membrane building, the uploaded coordinates are used to determine the volume of the molecular regions that would be located within, above, and below a membrane centered at $Z=0$ with a hydrophobic thickness of 24 Å.

<After>

To facilitate membrane building, the uploaded coordinates are used to determine the volume and center of mass of the molecular regions that would be located within, above, and below a membrane centered at $Z=0$ with a hydrophobic thickness of 24 Å.

STEP 2

<Before>

After dynamics, the CG models are replaced with all-atom models and collisions are minimized with up to 7 iterations of the greedy conformation search (see Supporting Information **Algorithm 1**).

<After>

Solvated components are initially positioned by applying a random rotation and aligning the component’s center of mass with the center of its corresponding CG particle. For membrane components, a root-mean-squared best-fit alignment is performed between the

resulting CG coordinates and the reference coordinates calculated in STEP 1; the Z axis rotation and X/Y translation results of this best fit are applied to the all-atom model. Then, collisions are minimized with up to 7 iterations of the greedy conformation search (see Supporting Information **Algorithm 1**).

[Comment 10] Protein-protein and protein-membrane contacts

- a. As the authors choose to reproduce already published results as validation, they should compare their results to those in Ref. 59.
- b. I am curious why in Fig 6 for the second row the probability for Ubiquitin-membrane interaction decreases at 10% and then increases at 30% (comparable to 0%). Since the last paragraph is very descriptive, I would appreciate an interpretation of the results also in light (and added value) to Ref. 59.

[Response 10] We added the following paragraph at the end of the section to address the Reviewer's comment.

<Added>

Although we modeled the same CHL1/POPC/PSM membrane and proteins as Nawrocki *et al.*,⁶¹ we observed overall higher protein aggregation. For example, their contact fraction between villin and other villin molecules (villin-villin contacts) ranges between 25-30% whereas ours is between 27-73%. Although we both report overall higher protein contact probabilities with increasing protein concentrations, there are exceptions to this trend: Nawrocki *et al.* show a statistically insignificant decrease in villin-villin contact between 5% and 10% v/v. In contrast, only our protein G-protein G contact fraction decreases between 5% and 10%. Notably, they used the CHARMM36 FF with modified ion parameters, and they also decreased protein aggregation by multiplying protein-water Lennard-Jones (LJ) potentials by 1.09, whereas we used the CHARMM36m FF that uses an updated CMAP potential to reduce left-handed α -helix formation but does not directly address protein-water interaction strength by default.⁴⁸ Due to these methodological differences, a direct comparison of our results with those of Nawrocki *et al.* is challenging. Further study could see if reintroducing water interaction scaling reconciles this discrepancy.

We further compared our protein-membrane interaction results with Nawrocki *et al.* in the second paragraph of the same section.

<Added>

Indeed, all protein-CHL1/POPC/PSM contacts surprisingly decrease between 5–10% v/v and increase between 10–30% v/v, suggesting a strong preference for protein-protein contacts that is only overcome by increased protein crowding. The higher protein-membrane contacts at 5% v/v can be explained by the lack of opportunities for protein cluster formation⁶¹. It should be noted that this trend has a small magnitude: even the highest protein-CHL1/POPC/PSM contact fraction (7.6% for ubiquitin-membrane at 30% v/v) is lower than the lowest protein-membrane contact fraction with other membranes (14.5% for protein G-axolemma at 5% v/v). We believe more simulation replicas are required to establish the statistical significance of this trend reversal. The overall low protein-CHL1/POPC/PSM contacts and high contacts with the other membranes we simulated are consistent with Nawrocki *et al.*'s finding that proteins are thermodynamically excluded from membranes with no charged lipids⁶¹.

[Comment 11] All the simulation details for all the different systems the authors use as validation should be described, including force-fields, simulation length, water models, ions etc. This is very important for the reproducibility study as the authors directly aim to compare to something.

[Response 11] We amended **Table S1** to include the caption below. Other simulation details—number of each component type, initial box dimensions, and simulation runtime—are included in this table.

Table S1 caption

<Added>

The TIP3P water model and 0.15 M KCl are used in all aqueous systems. All proteins and lipids are modeled with the CHARMM36(m) FF; polymers and CO₂ are modeled with CGenFF; mica and hydroxyapatite are modeled with the INTERFACE FF.

[Comment 12] Figure 7: it is unclear why the displacement in the bottom leaflet (D1) is slower. The authors also mention that the results are comparable to experiments but there is no mention of what type of experiments these are.

[Response 12] In all simulations containing Mica, the bottom POPC leaflet is the one closest to Mica. The difference between the systems is the thickness of the water layer separating Mica and POPC's bottom leaflet ($D_1 = 1$ nm, $D_2 = 2$ nm, $D_3 = 3$ nm). Thus, D_1 is the system that provides the greatest interaction between Mica and POPC. To quantify this interaction, we used the CHARMM `INTER` function to calculate interaction energies via the formula $\Delta E_{\text{inter}} = E_{\text{both}} - (E_{\text{POPC}} + E_{\text{Mica}})$ for the final frame of each simulation. CHARMM energy calculations automatically separate different energy terms; the VDW (ΔE_{VDW}) and electrostatic (ΔE_{elec}) terms are reported in **Table R1** below.

Table R1. Interaction energies between POPC and Mica in kcal/mol.

	$D_{1\text{ nm}}$	$D_{2\text{ nm}}$	$D_{3\text{ nm}}$
ΔE_{total}	-88.36	-4.65×10^{-3}	0
ΔE_{VDW}	-29.34	-3.95×10^{-3}	0
ΔE_{elec}	-59.01	-7.08×10^{-4}	0

We believe this attraction hinders lateral diffusion of POPC by increasing friction between POPC and water.

To mention what type of experiments were used in our comparison, we amended Results section 3 as shown below:

<before>

Experimental values of the POPC diffusion coefficient in a bilayer are $\sim 7 \mu\text{m}^2/\text{s}^{62}$, implying that a bilayer on support behaves like free-standing lipids when the water thickness is more than 2 nm. In contrast, when the thickness of the water layer is less than 2 nm, there are strong interactions between the support and the lipids, which reduces the diffusion coefficient of the lipid in the lower leaflet of the SLB. This finding is consistent with

experimental measurements, which have shown that the thickness of the water layer is in the range of 1~2 nm^{62,63}.

<after>

Experimental values of the POPC diffusion coefficient measured by raster image correlation spectroscopy (RICS) of a lipid-like probe in a GUV bilayer are $\sim 7 \pm 3 \mu\text{m}^2/\text{s}$ ⁶⁴, implying that a bilayer on support behaves like free-standing lipids when the water thickness is more than 2 nm. In contrast, when the thickness of the water layer is less than 2 nm, there are strong interactions between the support and the lipids, which reduces the diffusion coefficient of the lipid in the lower leaflet of the SLB. This finding is consistent with water thickness of dimyristoylphosphatidylcholine (DMPC) SLBs measured with specular reflection of neutrons, which found that water thickness is in the range of 2~4 nm⁶⁵.

References

64. Gielen, E. *et al.* Measuring Diffusion of Lipid-like Probes in Artificial and Natural Membranes by Raster Image Correlation Spectroscopy (RICS): Use of a Commercial Laser-Scanning Microscope with Analog Detection. *Langmuir* **25**, 5209–5218 (2009).
65. Johnson, S. J. *et al.* Structure of an adsorbed dimyristoylphosphatidylcholine bilayer measured with specular reflection of neutrons. *Biophysical Journal* **59**, 289–294 (1991).

[Comment 13] Diffusion of CO₂: More details on the system and comparison to existing data (simulation, experiment?) is required. The results are descriptive and lacking interpretation; hence it is unclear why these results are relevant and/or reproduce correctly the relevant data.

[Response 13] To address this comment, we added the following section to the beginning of Supplemental Information.

<Added>

CO₂ Diffusion

We used the methods in Im and Roux's 2002 study¹ to calculate position dependent diffusion coefficients along the Z axis ($D(z)$) for embedded CO₂ with lag times (τ) from 1–10 ns (**Figure S7**). Mean and standard errors were calculated from the same bin positions across different simulation replicas, and diffusion in the polymer center (**Table S4**) was calculated by averaging all observations within bins located between $-15 \text{ \AA} < z < 15 \text{ \AA}$.

We found that $D(z)$ varies substantially by Z position, even within the polymer center, as CO₂ molecules find defects in the polymer structure where local diffusion is increased on short time scales. The mean of diffusion across bins in the polymer center was calculated to be almost 2x larger for PET₉₅ ($7.8 \pm 2.2 \text{ cm}^2/\text{s} \times 10^{-8}$) than PEF₉₅ ($4.0 \pm 0.3 \text{ cm}^2/\text{s} \times 10^{-8}$), possibly due to there being more defects within the PET₉₅ structure. Of the studies of CO₂ diffusion in PET we surveyed (**Table S4**), the most similar rate is in a simulation ($2.43 \text{ cm}^2/\text{s} \times 10^{-7}$ at 25 °C) which used lag times on the order of picoseconds. An experimental study² reported much slower CO₂ diffusion ($7.2 \pm 2 \text{ cm}^2/\text{s} \times 10^{-11}$ and $2.2 \pm 0.4 \text{ cm}^2/\text{s} \times 10^{-9}$ for PEF and PET, respectively), however their finding that CO₂ diffuses more slowly through PEF than PET is consistent with ours. That study measured permeation more directly via changes in atmospheric pressure across a macroscopic plastic barrier, whereas our study measures diffusion by tracking individual particles in a microscopic barrier.

To compare with other diffusion studies, it should be noted that the scale of $D(z)$ depends on τ ; the smaller timescales of simulations necessitate smaller τ values, which tends to overestimate long-term diffusion compared to experiments. The utility of simulation studies in this domain is thus by comparison of relative diffusion rates, rather than calculation of absolute diffusion rates.

A new reference to the SI was added to the section “Diffusion of CO₂ through polymer membranes” in the main text.

<Before>

Notably, between 666–1333 ns, the CO₂ density in near the polymer center (0–12 Å) was higher for PET₉₅ than for PEF₉₅, and the CO₂ density near the polymer periphery (15–37 Å) was lower for PET₉₅ than for PEF₉₅ in the same time range, indicating that PEF₉₅ is overall more resistant to CO₂ diffusion.

<After>

Notably, between 666–1333 ns, the CO₂ density in near the polymer center (0–12 Å) was higher for PET₉₅ than for PEF₉₅, and the CO₂ density near the polymer periphery (15–37 Å) was lower for PET₉₅ than for PEF₉₅ in the same time range, indicating that PEF₉₅ is overall more resistant to CO₂ diffusion. Indeed, CO₂ diffusion measured in the polymer center was nearly twice as high for PET₉₅ ($7.8 \pm 2.2 \text{ cm}^2/\text{s} \times 10^{-8}$) versus PEF₉₅ ($4.0 \pm 0.3 \text{ cm}^2/\text{s} \times 10^{-8}$), as described in **Supplemental Information: CO₂ Diffusion**.

We added the following figure, table, and references to the Supplemental Information.

<Added>

Figure S7. Position dependent diffusion profiles of CO₂ through PET₉₅ and PEF₉₅ in its approximate center. The purple and green lines were calculated from the equation $D = \text{MSD}(\tau) / (2\tau)$ with lag times of 2 and 5 ns, where D is diffusion, $\text{MSD}(\tau)$ is the measured mean square displacement of CO₂ when using a given lag time τ . The blue lines result from linear regression of the same equation with all lag times (at 0.1 ns intervals) from 1.0 to 10.0 ns. Error bars represent the standard error of the mean across all simulation replicas within each bin.

Table S4. Selected diffusion constants in this and other studies. Error/confidence intervals are shown where included in the original study.

Reference	Study Type	Polymer	D_{CO_2} (cm ² /s)	Temperature
2	Experiment	PEF	$(7.2 \pm 2) \times 10^{-11}$	35 °C
2	Experiment	PET	$(2.2 \pm 0.4) \times 10^{-9}$	35 °C
3	Experiment	PET	1×10^{-9}	STP
4	Simulation	PET	2.43×10^{-7}	25 °C
This study	Simulation	PEF	$(1.7 \pm 0.1) \times 10^{-7}$	25 °C
This study	Simulation	PET	$(1.5 \pm 0.1) \times 10^{-7}$	25 °C

References

1. Im, W. & Roux, B. Ions and Counterions in a Biological Channel: A Molecular Dynamics Simulation of OmpF Porin from Escherichia coli in an Explicit Membrane with 1M KCl Aqueous Salt Solution. *J. Mol. Biol.* 319, 1177–1197 (2002).
2. Burgess, S. K., Kriegel, R. M. & Koros, W. J. Carbon Dioxide Sorption and Transport in Amorphous Poly(ethylene furanoate). *Macromolecules* 48, 2184–2193 (2015).
3. Okuji, S. et al. Surface modification of polymeric substrates by plasma-based ion implantation. *Nucl. Instrum. Methods Phys. Res. Sect. B Beam Interact. Mater. At.* 242, 353–356 (2006).
4. Liao, L.-Q., Fu, Y.-Z., Liang, X.-Y., Mei, L.-Y. & Liu, Y.-Q. Diffusion of CO₂ Molecules in Polyethylene Terephthalate/Poly lactide Blends Estimated by Molecular Dynamics Simulations. *Bull. Korean Chem. Soc.* 34, 753–758 (2013).

[Comment 14] The authors stress that the study presents an “automated and versatile procedure for building”. I recommend softening these types of statements as quite some manipulation is required, a fact also acknowledged by the authors in the next paragraph.

[Response 14] We reworded the conclusion to reflect the Reviewer’s comment.

<Before>

This work presents an automated and versatile procedure for building simulation-ready, atomistic models containing heterogeneous molecular components via *Multicomponent Assembler* (MCA) in CHARMM-GUI. Automation is achieved by assigning component types that determine the packing strategy and using a greedy packing algorithm whose initial configuration is generated from coarse-grained simulation.

<After>

This work presents a guided procedure for building simulation-ready, atomistic models containing heterogeneous molecular components via *Multicomponent Assembler* (MCA) in CHARMM-GUI. Initial positioning is facilitated by assigning component types that determine the packing strategy and using a greedy packing algorithm whose initial configuration is generated from CG simulation.

[Comment 15] General comments

- a. The figures would benefit from larger labels, thicker lines, and a better choice of colors. Generally, on a printed version of the manuscript most lines are not visible or indistinguishable, the labels and legends are too small.

- b. In Fig. S4 It would be useful to mention if these are representative structures and if so, how they were chosen. If clustering was performed, then how?

[Response 15] The structures in **S4** were identified by viewing the simulation trajectories in VMD. No clustering was performed. The structures were chosen only to provide an example of a long-lived contact event. The following statement was added to the **Figure S4** caption.

<Added>

These representative structures were identified visually.

We furthermore increased line thickness and font size in most figures. Thank you!

Reviewer #2 (Remarks on code availability):

The repository does not contain the actual code itself. The code that is presented in this paper is not publicly available and cannot be reviewed.

Analysis scripts are present, however, both coordinate input files as well as topology input files are not provided.

For reproducibility purposes, I strongly recommend that the authors provide these files. This seems especially important since the actual code cannot be reviewed. In my view, the editorial office should insist on this aspect unless very good reasons are given for not making this data publicly available.

Reviewer #3 (Remarks to the Author):

I positively value the responses and comments of Kern et al. in their manuscript, which now offers a more comprehensive view of their work. However, there are a few points that remained unaddressed, and which would make the manuscript suitable for publication.

[Comment 4] Each example should include a link to the demo as the reader has a hard time to identify the proper videos on Youtube. Particularly, it appears that not all examples in the paper have demos.

[Comment 5] I appreciate the inclusion of scripts on the Github but the concrete commands the user should follow are still not specified, which does not lower the entrance barrier.

[Comment 8] The authors did not reply to one of my questions in the sasa context, i.e. "why is achieving a volume fraction higher than 30% a trial-and-error process and how can this be avoided?"

[Comment 12] Response 12 should be included in the main text.

Reviewer 1

We thank the Reviewer again for the positive comments and checking our responses to questions from the other Reviewers. Since there are no suggestions to revise the manuscript, we have no specific responses.

Reviewer 2

We thank the Reviewer again for the positive comments and suggestions. We again revised the manuscript according to the Reviewer's comments as much as we could, and our specific responses are given below. To be clear, we use "black" for the Reviewer's current comments, "blue" for our current responses, and "dark green" for our previous responses.

[Comment to the previous Response R1]

The authors' argument is valid, which is why I considered this problem to be a minor point. I was not suggesting any comparison to Bentopy (indeed an unpublished package) or building the cell as a form of benchmark. I'm happy that the authors have included a more detailed benchmark comparison with PACKMOL, which is the most similar program capability-wise. Comparing the MCA capabilities to other programs is also a valuable addition. However, I've some comments on this addition (see below).

[Comment R1.1]

Moltemplate is designed for CG simulations, so that third-party molecular builder tools are required to construct all-atom models of biomolecules such as proteins, RNA, and DNA.

I'm a little confused by this sentence, since Moltemplate states on their website: "Moltemplate was intended for building custom coarse-grained molecular models, but it can be used to prepare realistic all-atom simulations as well." As far as material science systems such as polymer melts, surfaces, and solutions, are concerned, the moltemplate website lists quite some examples for generating these.

[Response R1.1]

According to the Reviewer's comment, we have amended the sentence as shown below.

<Before>

Moltemplate is designed for CG simulations, so that third-party molecular builder tools are required to construct all-atom models of biomolecules such as proteins, RNA, and DNA.

<After>

Although moltemplate was designed for custom CG modeling, it has been used in the preparation of all-atom models⁷². However, external tools are required to select appropriate FF atom types and resolve collisions in prepared models. According to their website, moltemplate "is not suitable for all-atom protein simulations".

[Comment R1.2]

As described in our recent publication³⁰, the initial polymer configurations generated by CHARMM-GUI *Polymer Builder* exhibit structures similar to fully relaxed configurations, allowing for direct production runs without additional equilibration simulations. However, when utilizing moltemplate's linear stacking method, additional equilibration simulation of tens to hundreds of nanoseconds are required even after the initial structure formation.

I find the above paragraph a little confusing, though I agree with the content. My suggestion would be to flip the two sentences around. Something along the lines of

"When utilizing moltemplate's linear stacking method to generate initial polymer conformations, long equilibration simulation of tens to hundreds of nanoseconds are required before the melt is well relaxed. In contrast, the CHARMM-GUI Polymer Builder creates initial structures similar to

fully relaxed configurations, allowing for direct production runs without long equilibration simulations.”

[Response R1.2]

This change sounds reasonable, and the paragraph has been updated accordingly.

[Comment R1.3]

Polyply has been developed for CG simulations using the Martini force field and recently extended to support all-atom models for certain long polymers and carbohydrates. However, similar to Moltemplate, third-party software is required for biomolecules.

This statement is inaccurate. Polyply has been developed to generate topologies and coordinates irrespective of the force field or resolution. Already the first published version supports all-atom and CG polymers. For example, the publication (<https://www.nature.com/articles/s41467-021-27627-4#Sec2>) contains benchmarks for polymer melts at the all-atom level as well as polymers at Martini resolution (see Figure 3b). Note that Table S2 should also be updated accordingly. According to the polyply GitHub page

(see: https://github.com/marrink-lab/polyply_1.0/blob/master/LIBRARY.md),

it currently supports the following forcefields: Martini 2/3, Amber variant ParmBsc1 for DNA, GROMOS (2016H66, and 53A6), OPLSAA. For the sake of not having unnecessarily long discussions of programs, I would propose to rephrase this paragraph along these lines of:

“Polyply has been developed to generate input files and starting coordinates for polymeric molecules independent of the level of resolution (e.g. Martini but also all-atom). However, it does not perform rigid body packing of large molecules (e.g. proteins) and requires third-party software for certain biomolecules, similar to Moltemplate.”

[Response R1.3]

We appreciate the detailed correction. In addition to the updated table, the statement has been amended as shown below.

<Before>

Polyply has been developed for CG simulations using the Martini force field and recently extended to support all-atom models for certain long polymers and carbohydrates. However, similar to Moltemplate, third-party software is required for biomolecules.

<After>

Polyply has been developed to generate input files and starting coordinates for polymeric molecules at CG and all-atom resolutions. However, it does not perform rigid body packing of large molecules and requires third-party software for certain biomolecules, similar to Moltemplate.

Table S2. [as amended, updates shown in black]

	MCA	Pysimm	Moltemplate	PACKMOL	Polyply
User Interface	GUI or CHARMM script	Python API	Special script	Special script	Special script

Supported Simulation Programs	GROMACS CHARMM OpenMM GENESIS Desmond	Amber	LAMMPS CASSANDRA	LAMMPS	N/A	GROMACS
Supported Force Fields	CHARMM AMBER		CHARMM AMBER DREIDING PCFF	GAFF MARTINI DREIDING	OPLS COMPASS MARTINI GAFF	N/A MARTINI GROMOS AMBER OPLS
Topology Preparation	Automatic		Automatic	Manual	N/A	Manual

[Comment R1.4]

See language edits in the part below.

To test whether our packing approach can lead to more dense packing and lower runtimes even **though** the CHARMM executable is not optimized for packing, we selected two sets of macromolecules shown in **Table S3**: one where all molecules have asphericity between 0.07 – 0.15 (easy) and the other where asphericity varies between 0.02 – 0.48 (hard). We then ran 12 replicas of both sets through PACKMOL and MCA with volume fraction (v/v) starting from 10% and increasing by 1% until all 12 replicas failed at the same v/v. As shown in **Figure S6**, MCA's runtimes were longer for **low-density** systems, and shorter for **high-density** systems. The maximum v/v achieved by PACKMOL was 30% (easy) and 18% (hard), and it was 41% (easy) and 23% (hard) for MCA. These results show that our approach can improve packing performance for periodic geometries when the system density is high.

[Response R1.4]

We have amended the language according to your suggestion.

[Comment to the previous Response R2]

The rephrased sentence is indeed more appropriate.

[Response R3]

Thank you!

[Comment to the previous Response R3]

MCA indeed does not re-implement the procedures in *Polymer Builder* or *Input Generator* (now *FF-Converter*), but instead builds on and integrates with them.

It doesn't practically integrate them, does it? The user still has to go to a separate builder module and work through the respective workflow (that's already published). Only then can the system be assembled. However, this statement was intended as an observation rather than a point of critique.

[Response R3]

Regarding *FF-Converter*, the format of assembled systems and their FFs are converted to the type expected by the user's chosen simulation programs without the need for a separate upload

to the *FF-Converter* module, as *Multicomponent Assembler* communicates with *FF-Converter* on behalf of the user.

For *Polymer Builder*, you are correct that the user must download and re-upload its output to *Multicomponent Assembler*. *Multicomponent Assembler* is designed to accept *Polymer Builder*'s PSF/CRD output with no modification besides (at most) a change in filenames. While this is somewhat of a hassle, it is less troublesome than trying to generate a CHARMM PSF/CRD from a PDB file containing polymers via *PDB Reader*.

[Comment to the previous Response R4]

As the Reviewer suggested, we performed the benchmark study shown in **Response 1** using MCA and PACKMOL using biomolecules with varying asphericity. The Syn3A model includes long chromosomes that cannot be modeled well by either MCA or PACKMOL. Therefore, we chose to instead model two smaller sets of representative molecules, as explained in **Response 1**.

This new benchmark nicely demonstrates the power of MCA. Consistently achieving nearly 30% packing density is a quite good result. I would recommend moving this benchmark to the main paper. However, in light of the already rather long main paper, this suggestion is optional.

[Response R4]

We are glad that the benchmark is viewed positively. Since we are concerned about the paper's length, we do prefer that the benchmark remains in the Supplemental Information section, unless the editor suggests otherwise. However, we agree that the result is important and should be emphasized better. We thus added the following short sentence in the main text:

At the end of **Workflow of Multicomponent Assembler** STEP 2 paragraph 2:

<Added>

As shown in our benchmark study (see **Figure S6** and **Comparison with other programs**), the difficulty of finding a collision-free packing depends on the shape of the components and the desired system density. In particular, for high volume fractions (e.g., > 30%), as shown in our benchmark testing, a trial-and-error process is unavoidable in the current approach, which can be improved with a better method in the future.

[Comment to the previous Response R5]

<Added>

The rigid body packing used by MCA or PACKMOL tends to work well for packing problems involving macromolecules that are already fairly rigid, such as proteins, crystals, and short nucleic acid sequences. Packing long flexible molecules such as synthetic and nucleic acid polymers is unlikely to succeed because the molecule's ability to pack tightly with other components comes from its flexibility. In such cases, the polymers should be assembled by another method, such as the random walk algorithms used by pysimm[**reference missing**] and polyply[**reference missing**] or the CG Kuhn fragment equilibration method used by *Polymer Builder*³⁰.

[Response R5]

The paragraph is updated to include the missing references.

[Comment to the previous Response R6]

The limitation section is a good addition to the paper. However, when reading the paragraph, I understood that I could create a polymer solution with polymer builder and then add the proteins using MCA. But from the general response, I gather that would likely not work, or would it? I'm insisting on this class of systems because they are quite common in bio-physics / bio-materials (e.g. hydrogels, coacervates, antibody formulations, joint fluids, and many more).

[Response R6]

You can add proteins to a system *containing* a polymer solution using MCA. You just can't easily add them to the same region of the system that contains the polymer. As you gathered, it is unlikely to succeed due to the high likelihood of randomly generated coordinates resulting in severe collisions that do not resolve with our rigid-body conformation search algorithm, unless there is a large amount of empty space between polymer molecules.

If the pre-arranged polymer solution is uploaded as a "periodic" component, then collisions with proteins can be avoided by preventing randomly initialized coordinates inside the box defined by the periodic component's XYZ dimensions specified by the user on the STEP 2 page. Clearly this was not adequately explained, so we added the following image to the Supplemental Information and new paragraph after paragraph 2 of Results and Discussion section 1 STEP 2:

<Added>

Systems without a membrane or periodic component are modeled with a cubic crystal lattice ($X=Y=Z$, 90° angles); those with a membrane are tetragonal ($X=Y\neq Z$, 90° angles) and require the system width (X or Y) and height (Z) to be determined separately (see **Figure S8A**); those with a periodic component are orthorhombic ($X\neq Y\neq Z$, 90° angles), with system X and Y dimensions defined by the periodic component's dimensions. For the purpose of estimating the available space for packing, the approximate membrane or periodic component's thickness along Z must be provided, if applicable (see **Figure S8B**). Furthermore, the provided membrane or periodic component's XYZ dimensions are reserved for that component. Note that their volumes are subtracted from the available space for system density calculations and solvent/solute components are prevented from being initialized within or entering the reserved regions.

Figure S8. Size determination options for systems with a membrane or periodic component. (A) The presence of a periodic component shows the periodic component size table, in which the user provides exact X and Y dimensions and an approximate Z dimension. If present, multiple periodic components must share X and Y dimensions. (B) The user can provide the exact length of the X dimension (X=Y) and estimate the membrane thickness along the Z axis. “Membrane thickness” is functionally equivalent to the periodic component Z length. (C) The X, Y, and Z lengths of any or periodic component membrane are used to reserve a box-shaped region for that component. Other components are initialized outside reserved regions and cannot enter them during packing.

[Comment to the previous Response R7]

We eventually plan to create a web API to facilitate automated access to CHARMM- GUI. However, it will be a big project because all CHARMM-GUI modules would need to be modified to support this type of access. We do not currently have any plans to connect MCA to *High Throughput Simulator*, but we will consider the idea if we get the necessary funding and talent.

Ok! I’m looking forward to when this is going to happen. See some language edits below on the added text section.

Although there is a large library of molecules that can be handled by MCA via other CHARMM-GUI modules, incorporating molecules that are not available in CHARMM-GUI—such as new membrane lipids, synthetic polymer building blocks, or ligands unsupported by CGenFF—could be **extremely [too informal]** challenging for users.

Although users can download the CHARMM scripts used to generate a system, the scripts are often hundreds to thousands of lines long. While MCA allows users to upload their

own CHARMM topologies (RTF) and parameters (PRM) for molecules not contained within the CHARMM36(m) or INTERFACE FFs, parameterizing molecules for the CHARMM FF is challenging. **However, many** other programs (e.g., psfgen for PSF⁷²; FFFParam¹², CGenFF¹⁶⁻¹⁸, ~~etc. for topologies and parameters; MDAnalysis⁷³ etc. for CRD~~) are capable of producing **these topology and parameter files.**

[MDAnalysis is not really a library for generating coordinates but rather converting or reading them; I would just mention the programs for topology and parameter files.]

[Response R7]

We have incorporated the reviewer's suggested changes.

[Comment to the previous Response R8]

The addition of run-time results nicely shows that MCA is faster than Packmol for most tasks. The overall run-time even for complex systems, is also more than acceptable. I'm satisfied with the discussion in the benchmark section.

[Response R8]

We are glad that the benchmark is again viewed positively.

[Comment to the previous Response R9]

Thank you for clarifying this point and editing the checklist!

However, after inspecting the mentioned repository more closely, I realized that simulation parameters and starting structures of the different test cases are not provided. To improve reproducibility and adhere to the FAIR principles of science, I need to insist that these be made available as well. If the authors are worried about file sizes, they can upload the data to for example Zenodo instead of GitHub.

[Response R9]

We expanded the GitHub repository to include (1) all initial simulation files (coordinates, settings, etc.) for each system presented in the system and (2) the script we used to run the simulation on our cluster.

Reviewer 3

We thank the Reviewer again for the positive comments and suggestions. We again revised the manuscript according to the Reviewer's comments as much as we could, and our specific responses are given below.

[Comment to the previous Response 4] Each example should include a link to the demo as the reader has a hard time to identify the proper videos on YouTube. Particularly, it appears that not all examples in the paper have demos.

[Response 4] Perhaps the easiest way to find them is to check the video demos page of CHARMM-GUI (<https://charmm-gui.org/demo>), particularly the "Multicomponent Assembler" submenu. However, the reviewer is correct that not all examples have demos; for the mica + POPC system, we added a description to the Supplemental Information, as described below.

To refer readers to the most relevant demo for each example, we added the following text to Methods section 1: Test System Preparation.

To "Solution systems with three proteins (5, 10, 30% v/v)":

<Added>

Preparation of a similar system is demonstrated in video demo "Solvating Multiple Proteins" (<https://charmm-gui.org/demo/multicomp/2>). To rebuild these systems, use the corresponding volume fraction and component counts shown in table S1 on the STEP 1 page.

To "Membrane systems with three proteins (5, 10, 30% v/v)":

<Added>

Preparation of a similar system is demonstrated in video demo "Solvating Proteins with Membranes and Membrane Proteins" (<https://charmm-gui.org/demo/multicomp/3>). To rebuild these systems, use the corresponding volume fraction and component counts shown in table S1 on the STEP 1 page, and instead of uploading 5O8F (as shown in the demo), check the box labeled "Generate a membrane for this system" to enable the membrane size options.

To "Pre-equilibrated axolemma membrane systems with three proteins (5, 10, 30% v/v)":

<Added>

Preparation of a similar system is demonstrated in the second example from video demo "Solvating Proteins with Membrane-like Polymers or Pre-Equilibrated Membranes" (<https://charmm-gui.org/demo/multicomp/4>) and from video demo "Building Nano-Bio Interface with Image Bonds" (<https://charmm-gui.org/demo/multicomp/5>). To rebuild these systems, use the corresponding volume fraction and component counts shown in table S1 on the STEP 1 page.

To "HAP with three proteins (5, 10, 30% v/v)":

<Added>

Preparation of a similar system is demonstrated in the second example from video demo "Solvating Proteins with Membrane-like Polymers or Pre-Equilibrated Membranes" (<https://charmm-gui.org/demo/multicomp/4>). To rebuild these systems, use the corresponding volume fraction and component counts shown in table S1 on the STEP 1 page.

To "Polymer EO₄₀EE₃₇ with three proteins (5, 10, 30% v/v)":

<Added>

Preparation of a similar system is demonstrated in the second example from video demo “Solvating Proteins with Membrane-like Polymers or Pre-Equilibrated Membranes” (<https://charmm-gui.org/demo/multicomp/4>). To rebuild these systems, use the corresponding volume fraction and component counts shown in table S1 on the STEP 1 page.

To “Diffusion of CO₂ through polymer membranes”:

<Added>

Preparation of the PET₉₅ example is demonstrated in the third example of video demo “Custom Solvent Composition, Gaseous Solvents” (<https://charmm-gui.org/demo/multicomp/6>). Preparation of the PEF₉₅ example is the same except that the PEF monomer is chosen during polymer generation.

To “Multi-layer system (mica + POPC membrane)”:

<Added>

The exact steps required to generate these examples in MCA are described in **Supplemental Information: Mica + POPC Generation**.

A new section was added to Supplemental Information:

<Added>

Mica + POPC Generation

In *Nanomaterial Modeler* (<https://charmm-gui.org/input/nanomaterial>), we selected Mica > Muscovite from the Nanomaterial Type menu. In Box Options, we entered the approximate lengths of X = 100, Y = 100, and Z = 30. *Nanomaterial Modeler* automatically rounds up to the nearest unit cell size for the selected material (displayed in parentheses), which in this case is X = 103.8 Å, Y = 108.2 Å, and Z = 30.1 Å.

After clicking "Next", it took about 2 minutes to build this mica model. On the next page, we ignored all options and selected "download .tgz" to obtain the mica-only model. We extracted the .tgz archive, located the files named `step1_nanomaterial.psf` and `step1_nanomaterial.crd`, and renamed them to `muscovite.psf` and `muscovite.crd`, respectively. We also used the values for A, B, and C in `step1_nanomaterial.str` as the reference values for *Multicomponent Assembler*.

In *Multicomponent Assembler*, we uploaded the muscovite PSF/CRD files, and clicked "Next". On the size determination page (STEP 1), we set the component type of muscovite to "Periodic". In the Periodic Component Size table, we used the values A and B saved in `step1_nanomaterial.str` for the length of X and Y, respectively (i.e., X = 103.836, Y = 108.1836) and estimated Z as ZMAX - ZMIN from the same file (i.e., Z = 29.92). We estimated the thickness of a pure POPC membrane as 46.86 and set the water thickness to 23.43. In the Periodic Components table, we clicked "Set Position" to leave space for a 10 Å thick layer of water between the bottom leaflet of POPC and muscovite by using the default constraint type and positioning type ("Fixed Z position" and "Center of mass", respectively) and set the component position to 48.33. After clicking "Calculate System Size" and "Next", we set the ratio of POPC to 1 in both leaflets, clicked "Show the system info", and clicked "Next". On the next page, we set the ion placing method to "Monte-Carlo" and used 0.15 M KCl and clicked "Next". On the Solvent Options page, we used default values, clicked "Calculate Solvent Composition", then clicked "Next". On the Input Options page, we selected "OpenMM" input generation, "NPT ensemble", and a temperature of 298.15 K. After input generation completed, we downloaded the result with "download .tgz".

We followed the procedure in the previous paragraph to create the Mica + POPC models with a 20 Å and 30 Å water gap, except that the component position of muscovite was set to 58.33 and 68.33, respectively. Since we did not use the default equilibration scheme, we made the following changes to each model's OpenMM input options. In `step6.3_equilibration.inp` through `step6.6_equilibration.inp`, we changed "pcouple" from "no" to "yes" and added the following lines:

```
p_ref    = 1.0, 1.0, 1.0 # Pressure (Pref or Pxx, Pyy, Pzz; bar)
p_type   = anisotropic   # MonteCarloBarostat type
p_scale  = XYZ
p_freq   = 100
```

Additionally, for `step6.6_equilibration.inp`, we increased the number of steps (nstep) from "250000" to "1250000".

Additionally, in writing this description, we noticed an error in our description of the mica model we obtained from *Nanomaterial Modeler*. We originally wrote that its length along the Z axis was 76.8 Å, but in fact 76.8 is the sum of the membrane thickness (46.9 Å) and mica's length along Z (29.9 Å). This has been corrected in the manuscript.

[Comment to the previous Response 5] I appreciate the inclusion of scripts on the Github but the concrete commands the user should follow are still not specified, which does not lower the entrance barrier.

[Response 5] As the reviewer requested, we added example usage to each section of the repository.

[Comment to the previous Response 8] The authors did not reply to one of my questions in the *sasa* context, i.e. "why is achieving a volume fraction higher than 30% a trial-and-error process and how can this be avoided?"

For high volume fractions (e.g., > 30%), as shown in our benchmark testing, a trial-and-error process is unavoidable and particularly dependent on factors like the system type and shape of components. We hope to devise a better packing method to avoid this in the future.

We added the following sentence at the end of the quoted paragraph:

<Added>

As shown in our benchmark study (see **Figure S6** and **Comparison with other programs**), the difficulty of finding a collision-free packing depends on the shape of the components and the desired system density. In particular, for high volume fractions (e.g., > 30%), as shown in our benchmark testing, a trial-and-error process is unavoidable in the current approach, which can be improved with a better method in the future.

[Comment to the previous Response 12] Response 12 should be included in the main text.

[Response 12] As the reviewer suggested, we moved the discussion and table to the end of the section "POPC diffusion near mica-supported lipid bilayers".

Reviewer #2 (Remarks to the Author):

All previous comments have been addressed. I can recommend publication as is. Nicely done!

Reviewer #2 (Remarks on code availability):

Scripts are available the actual source code is not.

Reviewer #3 (Remarks to the Author):

The authors have now addressed all my comments. Thank you and congratulations!

Reviewer #3 (Remarks on code availability):

The authors have now included a detailed description in the files.